# Transcriptional regulation of neural stem cell expansion in the adult hippocampus

Nannan Guo[1,2,3,4‡], Kelsey D McDermott[5†], Yu-Tzu Shih[1,2,3,4†], Haley Zanga[1,2,3,4], Debolina Ghosh[1,2], Charlotte Herber[1,2], William R Meara[1,2], James Coleman[1,2], Alexia Zagouras[1,2], Lai Ping Wong[6], Ruslan Sadreyev[6], J Tiago Gonçalves[5], Amar Sahay[1,2,3,4]*

[1]Center for Regenerative Medicine, Massachusetts General Hospital, Boston, United States; [2]Harvard Stem Cell Institute, Cambridge, United States; [3]Department of Psychiatry, Massachusetts General Hospital, Harvard Medical School, Boston, United States; [4]BROAD Institute of Harvard and MIT, Cambridge, United States; [5]Ruth L. and David S. Gottesman Institute for Stem Cell Biology and Regenerative Medicine; Dominick Purpura Department of Neuroscience, Albert Einstein College of Medicine, Bronx, United States; [6]Department of Molecular Biology, Massachusetts General Hospital, Harvard Medical School, Boston, United States

**\*For correspondence:**
asahay@mgh.harvard.edu

[†]These authors contributed equally to this work

**Present address:** [‡]Key Laboratory of Mental Health of the Ministry of Education, Guangdong-Hong Kong-Macao Greater Bay Area Center for Brain Science and Brain-Inspired Intelligence, Guangdong Province Key Laboratory of Psychiatric Disorders, Department of Neurobiology, School of Basic Medical Sciences, Southern Medical University, Guangzhou, China

**Competing interest:** The authors declare that no competing interests exist.

**Abstract** Experience governs neurogenesis from radial-glial neural stem cells (RGLs) in the adult hippocampus to support memory. Transcription factors (TFs) in RGLs integrate physiological signals to dictate self-renewal division mode. Whereas asymmetric RGL divisions drive neurogenesis during favorable conditions, symmetric divisions prevent premature neurogenesis while amplifying RGLs to anticipate future neurogenic demands. The identities of TFs regulating RGL symmetric self-renewal, unlike those that regulate RGL asymmetric self-renewal, are not known. Here, we show in mice that the TF Kruppel-like factor 9 (*Klf9*) is elevated in quiescent RGLs and inducible, deletion of *Klf9* promotes RGL activation state. Clonal analysis and longitudinal intravital two-photon imaging directly demonstrate that Klf9 functions as a brake on RGL symmetric self-renewal. In vivo translational profiling of RGLs lacking Klf9 generated a molecular blueprint for RGL symmetric self-renewal that was characterized by upregulation of genetic programs underlying Notch and mitogen signaling, cell cycle, fatty acid oxidation, and lipogenesis. Together, these observations identify Klf9 as a transcriptional regulator of neural stem cell expansion in the adult hippocampus.

## Editor's evaluation

In this study, Guo et al. uncover a role for the transcription factor Klf9 in keeping adult hippocampal neural stem cells in a state of quiescence. Following relief from this molecular brake through Klf9 loss-of-function, neural stem cells undergo symmetric cell divisions that promote their self-renewal and expansion. This data suggest that Klf9 contributes to the molecular interplay that governs stem cell decisions between quiescence and activation on one hand and between distinct modes of cell divisions on the other.

## Introduction

In the adult mammalian brain, radial-glial neural stem cells (RGLs) in the dentate gyrus subregion of the hippocampus give rise to dentate granule cells and astrocytes (*Seri et al., 2001*; *Garcia et al., 2004*; *Ahn and Joyner, 2005*; *Lagace et al., 2007*; *Bonaguidi et al., 2011*; *Encinas et al., 2011*; *Gonçalves et al., 2016b*; *Moss et al., 2016*; *Pilz et al., 2018*), a process referred to as adult hippocampal

**eLife digest** In humans and other mammals, a region of the brain known as the hippocampus plays important roles in memory. New experiences guide cells in the hippocampus known as radial-glial neural stem cells (RGLs) to divide to make new neurons and other types of cells involved in forming memories.

Each time an RGL divides, it can choose to divide asymmetrically to maintain a copy of itself and make a new cell of another type, or divide symmetrically (a process known as symmetric self-renewal) to produce two RGLs. Symmetric self-renewal helps to restore and replenish the pool of stem cells in the hippocampus that are lost due to injury or age, allowing us to continue making new neurons.

Proteins known as transcription factors are believed to control how RGLs divide. Previous studies have identified several transcription factors that regulate the RGLs splitting asymmetrically to make neurons and other cells. But the identities of the transcription factors that regulate symmetric self-renewal in the adult hippocampus have remained elusive.

Here, Guo et al. searched for transcription factors that regulate symmetric self-renewal of RGLs in mice. The experiments found that RGLs that are resting and not dividing (referred to as 'quiescent') have higher levels of a transcription factor called Klf9 than RGLs that are actively dividing. Loss of the gene encoding Klf9 triggered quiescent RGLs to start dividing, and further experiments showed that Klf9 directly inhibited symmetric self-renewal. Guo et al. then used an approach called in vivo translational profiling to generate a blueprint that revealed new insights into the molecular processes involved in this symmetric division.

These findings pave the way for researchers to develop strategies that may expand the numbers of stem cells in the hippocampus. This could eventually be used to help replenish brain circuits with neurons and improve the memory of individuals with Alzheimer's disease or other conditions that cause memory loss.

neurogenesis (*Altman and Das, 1965*; *Eriksson et al., 1998*; *Spalding et al., 2013*; *Boldrini et al., 2018*; *Sorrells et al., 2018*; *Moreno-Jiménez et al., 2019*; *Tobin et al., 2019*; *Gage, 2019*; *Knoth et al., 2010*). Adult-born dentate granule cells integrate into hippocampal circuitry by remodeling the network and ultimately contribute to hippocampal-dependent learning and memory and regulation of emotion (*Gonçalves et al., 2016b*; *Anacker and Hen, 2017*; *Miller and Sahay, 2019*). Levels of adult hippocampal neurogenesis are highly sensitive to experience (*Cope and Gould, 2019*; *Vicidomini et al., 2020*) suggesting that neurogenesis may represent an adaptive mechanism by which hippocampal-dependent memory functions are optimized in response to environmental demands. Essential to this adaptive flexibility is the capacity of RGLs to balance long-term maintenance with current or future demands for neurogenesis ('anticipatory neurogenesis') in response to distinct physiological signals (*Bonaguidi et al., 2011*; *Cope and Gould, 2019*; *Vicidomini et al., 2020*; *Dranovsky et al., 2011*; *Schouten et al., 2020*).

Depending on environmental conditions, RGLs make decisions to stay quiescent or self-renew asymmetrically or symmetrically. Whereas enriching experiences (e.g., complex environments, exploration, and socialization) bias RGLs toward asymmetric divisions to generate astrocytes and neurons (*Dranovsky et al., 2011*; *Song et al., 2012*), unfavorable conditions promote RGL quiescence (e.g., chronic stress and aging) or symmetric self-renewal to support neural stem cell (NSC) expansion at the expense of neurogenesis (e.g., social isolation, seizures, and aging) (*Dranovsky et al., 2011*; *Sierra et al., 2015*; *Ibrayeva et al., 2021*). Asymmetric self-renewal of RGLs predominates over symmetric self-renewal division mode in the adult hippocampus and it ensures maintenance of RGL numbers while supporting current neurogenic demands (*Pilz et al., 2018*; *Vicidomini et al., 2020*). Conversely, symmetric self-renewal decouples RGL divisions from differentiation and is thought to serve distinct functions. First, symmetric divisions prevent premature differentiation of RGLs in a nonpermissive or unhealthy niche, and consequently, avert aberrant integration of adult-born dentate granule cells detrimental to hippocampal functions (*Ibrayeva et al., 2021*; *Cho et al., 2015*). As such, RGL amplification anticipates future demands for neurogenesis upon return to favorable conditions. Second, RGL expansion may represent an efficient mechanism to replenish the adult RGL pool after injury. Third, symmetric stem cell divisions maybe more efficient than asymmetric divisions for long-term

maintenance since fewer divisions are required to maintain RGL numbers. Furthermore, symmetric divisions may be associated with a lower rate of mutations and reduced replicative aging (*Shahriyari and Komarova, 2013*).

Extracellular physiological signals recruit transcription factors (TFs) within adult hippocampal RGLs to execute quiescence-activation decisions and symmetric or asymmetric self-renewal divisions (*Vici-domini et al., 2020*; *Andersen et al., 2014*; *Urbán et al., 2019*). A growing number of transcriptional regulators of quiescence and asymmetric (neurogenic or astrogenic) stem cell renewal have been identified (*Mukherjee et al., 2016*; *Jones et al., 2015*; *Zhang et al., 2019*; *Ehm et al., 2010*; *Imayoshi et al., 2010*). Deletion of such factors results in loss of RGL quiescence, increased neurogenesis and ultimately, differentiation-coupled depletion of the RGL pool. In sharp contrast, the identities of TFs that regulate RGL expansion have remained elusive. Here, we report that expression of the ubiquitously expressed TF, Kruppel-like factor 9 (*Klf9*), a regulator of dendritic and axonal plasticity in postmitotic neurons (*Moore et al., 2009*; *McAvoy et al., 2016*), is elevated in nondividing RGLs compared to dividing RGLs. Inducible genetic upregulation of *Klf9* in RGLs and progenitors decreased activation, whereas conditional cell-autonomous deletion of *Klf9* in RGLs promoted an activated state. Clonal lineage tracing and longitudinal two-photon imaging of adult hippocampal RGLs in vivo directly demonstrated a role for *Klf9* as a brake on symmetric self-renewal. In vivo translational profiling of RGLs generated a molecular blueprint for RGL expansion in the adult hippocampus: we found that loss of Klf9 in RGLs results in downregulation of a program of quiescence-associated factors and upregulation of genetic (mitogen, notch) and metabolic (fatty acid oxidation and lipid signaling) programs underlying RGL symmetric self-renewal. Together, these data identify Klf9 as a transcriptional regulator of NSC expansion in the adult hippocampus. Our study contributes to an emerging framework for how experiential signals may toggle a balance of transcriptional regulators of symmetric and asymmetric self-renewal of RGLs to amplify NSCs or asymmetrically divide and generate neurons and astrocytes.

## Results

### Inducible *Klf9* loss promotes RGL activation

To characterize *Klf9* expression in RGLs in the adult dentate gyrus, we bred *Klf9-LacZ* knockin reporter mice (*Scobie et al., 2009*) with a Nestin GFP transgenic mouse line in which Nestin+ RGLs are genetically labeled with GFP (*Mignone et al., 2004*). Quantification of *Klf9* expression based on LacZ intensities in *Klf9 $^{LacZ/+}$* mice revealed enrichment in quiescent RGLs relative to activated RGLs (MCM2+) (one-way analysis of variance [ANOVA], $F = 17.07$, $p = 0.003$) (*Figure 1A, B*). MCM2 expression captures activated cells that have exited quiescence. To refine this estimation that is based on a surrogate (LacZ) of *Klf9* expression within the RGL compartment, we performed fluorescence in situ hybridization (FISH) using a *Klf9*-specific riboprobe and immunohistochemistry for GFP and BrdU on adult hippocampal sections obtained from *Klf9 $^{+/+ or LacZ/LacZ}$*;Nestin GFP transgenic mice perfused 2 hr following a BrdU pulse (one-way ANOVA, $F = 5.6$, $p = 0.04$) (*Figure 1C–E*). No signal was detected with FISH using the *Klf9* riboprobe on brain sections from Klf9 $^{LacZ/LacZ}$ mice thus conveying specificity of the riboprobe (*Figure 1D*). Quantification of *Klf9* transcripts using Image J revealed significantly enriched expression in quiescent vs. activated (Brdu+ Nestin GFP+) RGLs (*Figure 1C, E*).

We next asked what happens when we delete *Klf9* in adult hippocampal RGLs. To address this question, we engineered *Klf9* conditional mutant mice (*Klf9 $^{f/f}$*) to cell autonomously delete *Klf9* in RGLs. We first validated our *Klf9 $^{f/f}$* mouse line by crossing it with the *POMC*-Cre mouse line that drives recombination in the dentate gyrus. In situ hybridization (ISH) on hippocampal sections from *POMC*-Cre:*Klf9 $^{f/f}$* revealed salt and pepper expression of *Klf9* in the dentate gyrus consistent with the established pattern of *POMC*-Cre-dependent recombination in the dentate gyrus (*McHugh et al., 2007*). No signal was detected by ISH using the *Klf9* riboprobe on brain sections from *Klf9 $^{LacZ/LacZ}$* mice thus conveying specificity of the riboprobe (*Figure 1—figure supplement 1*). We bred *Klf9 $^{f/f}$* mice with (GLI-Kruppel family member 1) *Gli1 $^{CreERT2}$* to recombine *Klf9* (deletion of exon 1) in hippocampal RGLs (*Figure 1F, G*). We chose the *Gli1 $^{CreERT2}$* driver line because population-based lineage tracing and chronic in vivo imaging suggests that *Gli1 $^{CreERT2}$*-labeled RGLs contribute to long-term maintenance and self-renewal (*Ahn and Joyner, 2005*; *Bottes et al., 2021*). We next generated *Klf9 $^{f/f or +/+}$* mice harboring a *Gli1 $^{CreERT2}$* allele and a Cre-reporter allele (Ai14, B6;129S6-*Gt(ROSA)26Sor $^{tm14(CAG-tdTomato)}$*

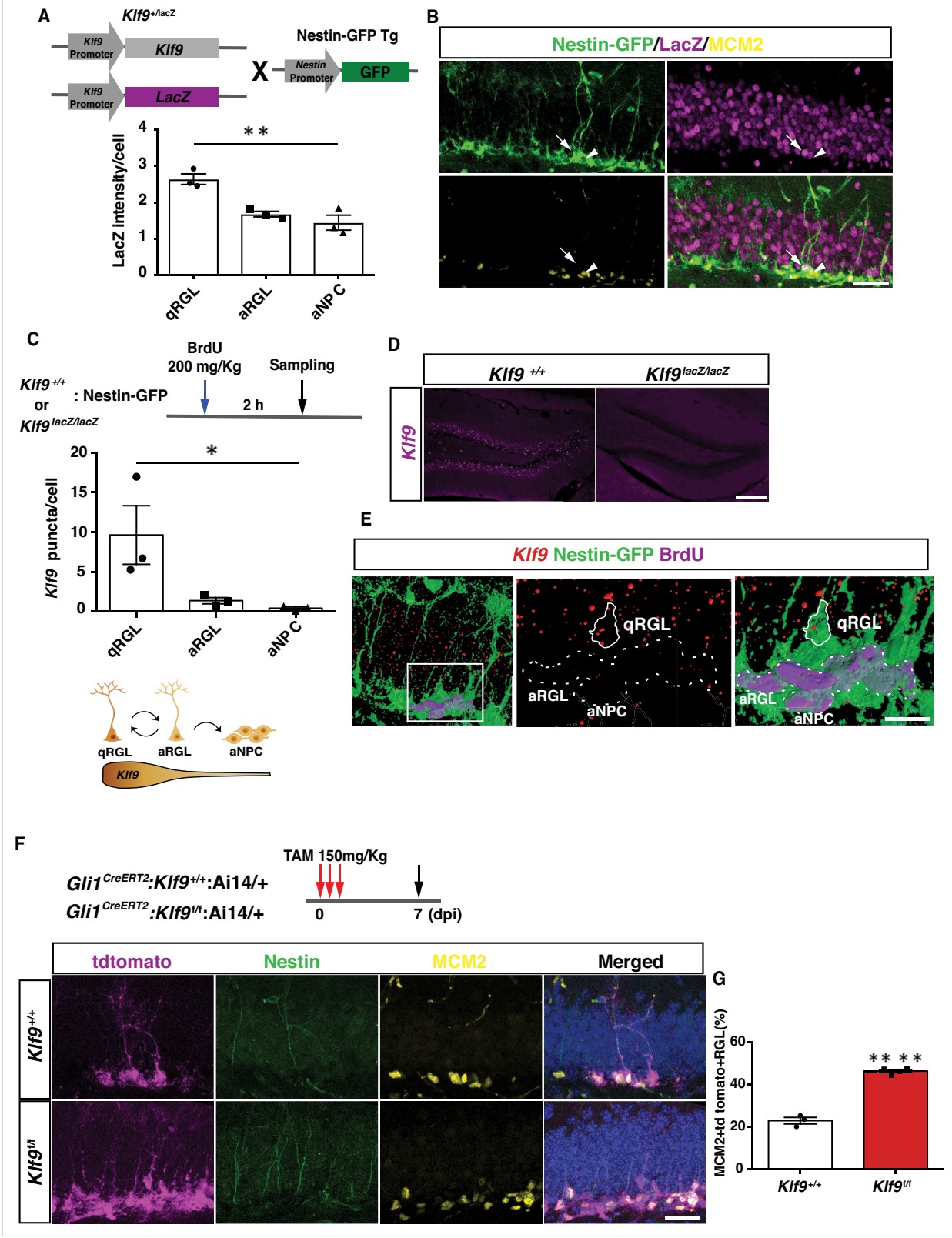

**Figure 1.** *Klf9* is elevated in nondividing radial-glial neural stem cells (RGLs) and loss of Kruppel-like factor 9 (Klf9) promotes RGL activation. (**A, B**) *Klf9* expression inferred from LacZ expression intensity in quiescent RGLs, qRGL (GFP+ MCM2 with radial process, arrows), activated RGL, aRGL (GFP+ MCM2+ with radial process, arrowheads) and activated neural progenitors, aNPCs (GFP+ MCM2+ without a radial process) in *Klf9*^LacZ/+;Nestin GFP transgenic. qRGLs exhibit higher *Klf9* expression than aRGLs and aNPCs. *n* = 3 mice/group. (**C–E**) Fluorescence in situ hybridization using a *Klf9*-specific

*Figure 1 continued on next page*

*Figure 1 continued*

riboprobe and immunohistochemistry for GFP and BrdU on adult hippocampal sections obtained from *Klf9^LacZ/+ or LacZ/LacZ*;Nestin GFP transgenic mice. (**D**) Specificity of riboprobe established by detection of *Klf9* expression in dentate gyrus of *Klf9^+/+* but not in *Klf9^LacZ/LacZ* mice. (**C, E**) *Klf9* is expressed in qRGLs but not in dividing (BrdU+) RGLs or aNPCs. *n* = 3 mice/group. (**F, G**) Inducible deletion of *Klf9* in Gli1+ RGLs in adult mice (*Gli1^CreERT2*:*Klf9^+/+*:Ai14 vs. *Gli1^CreERT2*:*Klf9^f/f*:Ai14) results in increased RGL activation (percentage of MCM2+ tdTomato + Nestin+ RGLs). *n* = 3 and 4 mice/group. Data are represented as mean ± standard error of the mean (SEM). *p < 0.05, **p < 0.01, ****p < 0.0001. Scale bar: **B, F**, 50 μm; **D**, 250 μm; **E**, 20 μm.

The online version of this article includes the following figure supplement(s) for figure 1:

**Figure supplement 1.** Generation and characterization of Kruppel-like factor 9 (*Klf9*) conditional mutant mouse line.

**Figure supplement 2.** Estimation of Kruppel-like factor 9 (*Klf9*) recombination frequency in Gli1-positive tdTomato-labeled radial-glial neural stem cells (RGLs).

**Figure supplement 3.** Inducible overexpression of Kruppel-like factor 9 (*Klf9*) in activated neural stem and progenitors promotes quiescence.

---

*^Hze*/J) (*Madisen et al., 2010*) to indelibly label Gli1-positive RGLs and their progeny (*Figure 1F*). By analysis of *Klf9* transcript-associated fluorescence intensity in Gli1-positive tdTomato-labeled RGLs we estimated the recombination frequency of *Klf9* (i.e., reduction in *Klf9*-associated signal) to be approximately 32% in Gli1-positive tdTomato-labeled RGLs (*Figure 1—figure supplement 2*). We induced *Klf9* recombination and tdTomato expression in RGLs of adult (2 months old) *Gli1^CreERT2*;*Klf9^f/f or +/+*; Ai14 mice and processed brain sections for Nestin, tdTomato, and MCM2 immunohistochemistry 7 days postinjection (7 dpi) to quantify activated RGLs (*Figure 1F, G*). We found that conditional deletion of *Klf9* in *Gli1^CreERT2*-targeted adult hippocampal RGLs significantly increased the fraction of activated RGLs (% of MCM2+ tdTomato + RGLs) (*Figure 1G*; unpaired *t*-tests, *Figure 1G*, p < 0.0001). Complementing these results, we demonstrated that genetic overexpression of *Klf9* in activated hippocampal NSCs and progenitors of adult *Sox1 tTA*;*tet0 Klf9* mice (*McAvoy et al., 2016*; *Venere et al., 2012*) significantly decreased the fraction of activated and dividing cells (*Figure 1—figure supplement 3*). Together, these data demonstrate that *Klf9* expression is enriched in quiescent RGLs and that loss of *Klf9* expression in RGLs either promotes or maintains an activated state of RGLs in the adult hippocampus.

## *Klf9* deletion in RGLs produces supernumerary RGL clones

We next asked how Klf9 loss-of-function in RGLs affects self-renewal division mode. Population-level lineage tracing experiments at short-term chase time points suggested that Klf9 loss in Gli1+ RGLs increased RGL numbers (data not shown). However, analysis of NSC dynamics at the population level is encumbered by changes in numbers of labeled progeny overtime (*Bonaguidi et al., 2011*; *Bottes et al., 2021*). The challenges of interpreting population-level analysis are exacerbated because *Klf9* is also expressed in immature adult-born neurons and mature dentate granule cells. As such, changes in numbers of labeled descendants following loss of Klf9 in RGLs make population-level lineage tracing difficult to interpret. Therefore, to directly investigate whether loss of *Klf9* in RGLs results in NSC expansion at a single clone level, we performed in vivo clonal analysis in adult *Gli1^CreERT2*;*Klf9^f/f or +/+*; Ai14 mice shortly after low-dose tamoxifen adminsitration. Single dose of TAM at 50 mg/kg body weight permitted sparse labeling of single tdTomato+ RGLs and visualization of labeled single RGL clones and their individual constituents. Analysis of clonal composition at 7 dpi revealed a significantly greater fraction of multi-RGL containing clones and a smaller fraction of single RGL containing clones (*Figure 2A–D*, *Figure 2—figure supplement 1*, *Figure 2—videos 1–8*; *Figure 2B*, two-way ANOVA, genotype × cell type p < 0.0001, Bonferroni post hoc *Klf9^+/+* vs. *Klf9^f/f.* 1 RGL n.s., 2+ RGLs p < 0.0001, 1 RGL+ p < 0.0001, no RGL n.s. *Figure 2D*, left panel, two-way ANOVA, genotype × cell type p < 0.0001, Bonferroni post hoc *Klf9^+/+* vs. *Klf9^f/f.* 2+ RGLs p = 0.09, 2 RGLs+ P + A p < 0.0001, 2 RGLs+ p n.s. *Figure 2D*, right panel, two-way ANOVA, genotype × cell type p = 0.09, Bonferroni post hoc *Klf9^+/+* vs. *Klf9^f/f.* 1 RGL n.s., 1 RGL+ P + A p = 0.01, 1 RGL+ P p = 0.01). Many of the multi-RGL containing clones also comprised of neural progenitors and astrocytes suggesting that loss of *Klf9* biases RGL expansion but does not prevent RGL differentiation into progeny (*Figure 2D*). To corroborate these findings and address any potential confound introduced by bias in the Ai14 genetic lineage tracer, we performed clonal analysis at 7 dpi using a different lineage tracing reporter, mTmG (*t(ROSA)26Sor^tm4(ACTB-tdTomato,-EGFP)Luo*/J) (*Muzumdar et al., 2007*). Our analysis demonstrated a significant increase in multi-RGL clones and decrease in single RGL clones derived from RGLs lacking

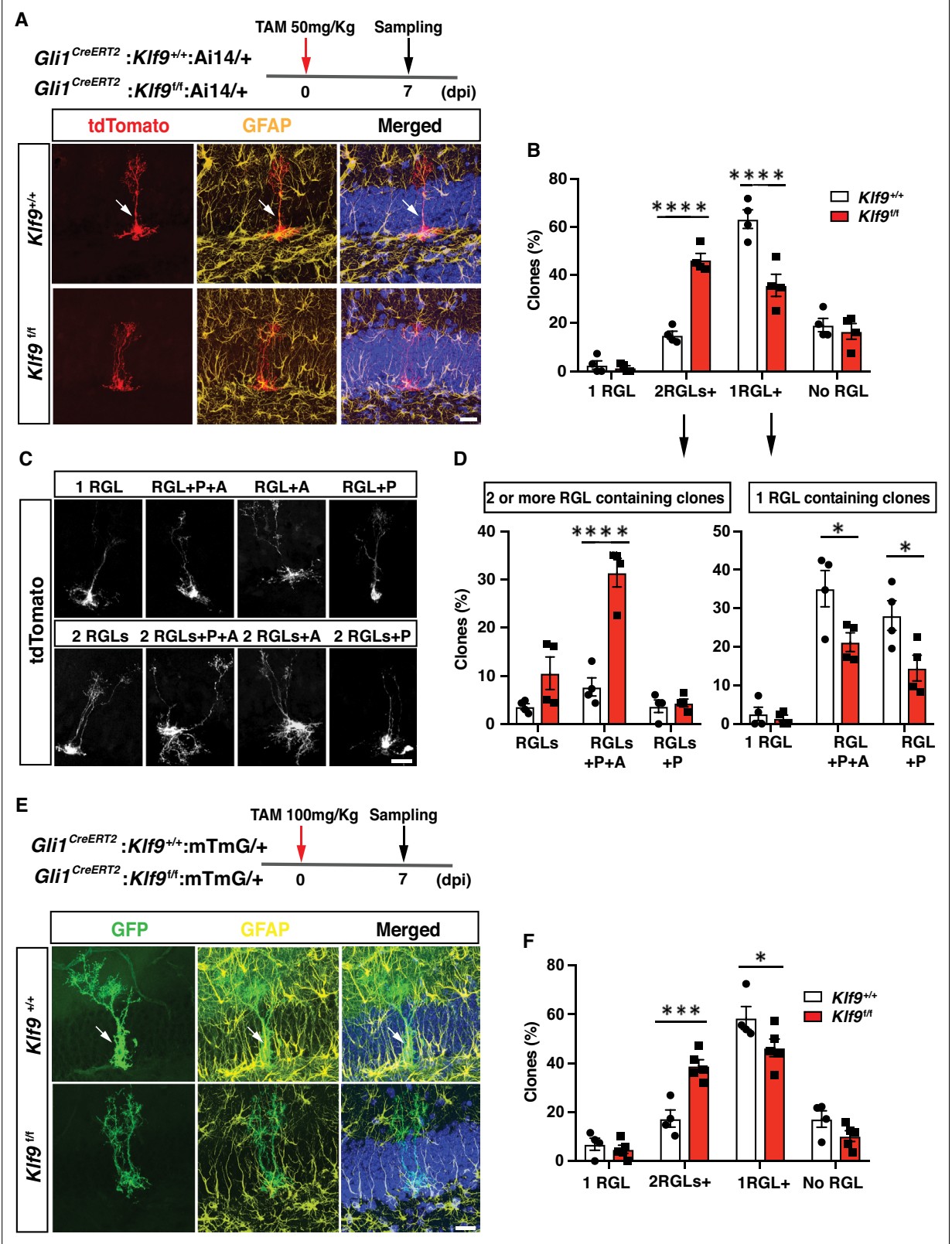

**Figure 2.** Kruppel-like factor 9 (*Klf9*) deletion in radial-glial neural stem cells (RGLs) produces supernumerary RGL clones. (**A–D**) Clonal analysis of sparsely labeled Gli1+ RGLs in adult *Gli1^CreERT2^:Klf9^+/+or f/f^*:Ai14 mice at 7 dpi. (**A, C**) Representative images of labeled RGL clones and descendants. For example, A top: single RGL (white arrow), A bottom: 2 RGLs. Identification was based on tdTomato+ morphology and GFAP immunohistochemistry. (**B**) Statistical representation of clones for specified compositions for both genotypes expressed as fraction of total clones quantified. (**D**) Breakdown

*Figure 2 continued on next page*

*Figure 2 continued*

of clones into 2 RGLs+ (two or more RGL containing clones and progeny) and single RGL+ clones (clones containing only 1 RGL and progeny). Loss of Klf9 in Gli1+ RGLs results in statistically significant overrepresentation of two or more RGL containing clones and significant reduction in '1 RGL containing clones' suggestive of *Klf9* repressing RGL expansion. *n* = 4 mice/group. (**E, F**) Clonal analysis of sparsely labeled Gli1+ RGLs (white arrow) in adult *Gli1^CreERT2:Klf9^+/+or f/f*:mTmG mice at 7 dpi. Inducible deletion of *Klf9* in Gli1+ RGLs results in statistically significant overrepresentation of multi-RGL containing clones (two or more RGLs, 2 RGLs+) and a significant reduction in single RGL containing clones (1 RGL+). Identification was based on GFP+ morphology and GFAP immunohistochemistry. Representative images (**E**) and corresponding quantification in (**F**). *n* = 4 and 5 mice/group. P: rogenitor(s), A: astrocyte. Data are represented as mean ± standard error of the mean (SEM). *p < 0.05, ***p < 0.001, ****p < 0.0001. Scale bar: **A, C, E**, 20 μm.

The online version of this article includes the following video and figure supplement(s) for figure 2:

**Figure supplement 1.** Analysis of clonal composition in *Figure 2C*.

**Figure 2—video 1.** Three-dimensional (3D) images of representative radial-glial neural stem cell (RGL) clonal compositions depicted in Figure 2C 1 RGL.

https://elifesciences.org/articles/72195/figures#fig2video1

**Figure 2—video 2.** Three-dimensional (3D) images of representative radial-glial neural stem cell (RGL) clonal compositions depicted in Figure 2C 1 RGL.

https://elifesciences.org/articles/72195/figures#fig2video2

**Figure 2—video 3.** Three-dimensional (3D) images of representative radial-glial neural stem cell (RGL) clonal compositions depicted in Figure 2C 1 RGL+ P + A.

https://elifesciences.org/articles/72195/figures#fig2video3

**Figure 2—video 4.** Three-dimensional (3D) images of representative radial-glial neural stem cell (RGL) clonal compositions depicted in Figure 2C 1 RGL+ A.

https://elifesciences.org/articles/72195/figures#fig2video4

**Figure 2—video 5.** Three-dimensional (3D) images of representative radial-glial neural stem cell (RGL) clonal compositions depicted in Figure 2C 1 RGL+ P.

https://elifesciences.org/articles/72195/figures#fig2video5

**Figure 2—video 6.** Three-dimensional (3D) images of representative radial-glial neural stem cell (RGL) clonal compositions depicted in Figure 2C 2 RGLs.

https://elifesciences.org/articles/72195/figures#fig2video6

**Figure 2—video 7.** Three-dimensional (3D) images of representative radial-glial neural stem cell (RGL) clonal compositions depicted in Figure 2C 2 RGLs+ P + A.

https://elifesciences.org/articles/72195/figures#fig2video7

**Figure 2—video 8.** Three-dimensional (3D) images of representative radial-glial neural stem cell (RGL) clonal compositions depicted in Figure 2C 2 RGLs+ P.

https://elifesciences.org/articles/72195/figures#fig2video8

*Klf9* in *Gli1^CreERT2;Klf9 ^f/f*mTmG mice (*Figure 2E, F*; *Figure 2F*, two-way ANOVA, genotype × cell type p < 0.0001, Bonferroni post hoc *Klf9^+/+* vs. *Klf9^f/f.* 1 RGL n.s., 2 RGLs+ p = 0.0001, 1 RGL+ p = 0.03, no RGLs n.s). The lack of a difference in single RGL clones suggests that loss of Klf9 maintains the activated state associated with increased symmetric divisions. Alternatively, it may reflect a floor effect (estimation of recombination frequency suggests that not all tdTomato RGLs undergo recombination for Klf9) in the assay that occludes detection of a decrease in number. These findings provide evidence for *Klf9* in cell-autonomous regulation of RGL expansion and are *suggestive* of a role for Klf9 in inhibition of symmetric self-renewal of RGLs.

### *Klf9* functions as a brake on symmetric self-renewal of RGLs

To unequivocally establish clonal origin of labeled progeny and directly test the hypothesis that *Klf9* inhibits symmetric self-renewal of RGLs in vivo, we performed longitudinal two-photon imaging (*Gonçalves et al., 2016a*) of RGLs for up to 2 months and tracked symmetric and asymmetric division patterns (*Figure 3A, B*; *Figure 3—figure supplement 1*, *Figure 3—videos 1–4*). We implanted *Gli1^CreERT2;Klf9^f/f or +/+*;Ai14 mice with a hippocampal window over CA1 for long-term imaging. After allowing 2 weeks for recovery from surgery, we injected mice with a single dose of tamoxifen (150 mg/kg) to induce Cre recombination and tdTomato expression in Gli1+ RGLs (as shown in *Figure 1—figure supplement 2*). This resulted in sparse labeling that allowed us to image and track individual cells and

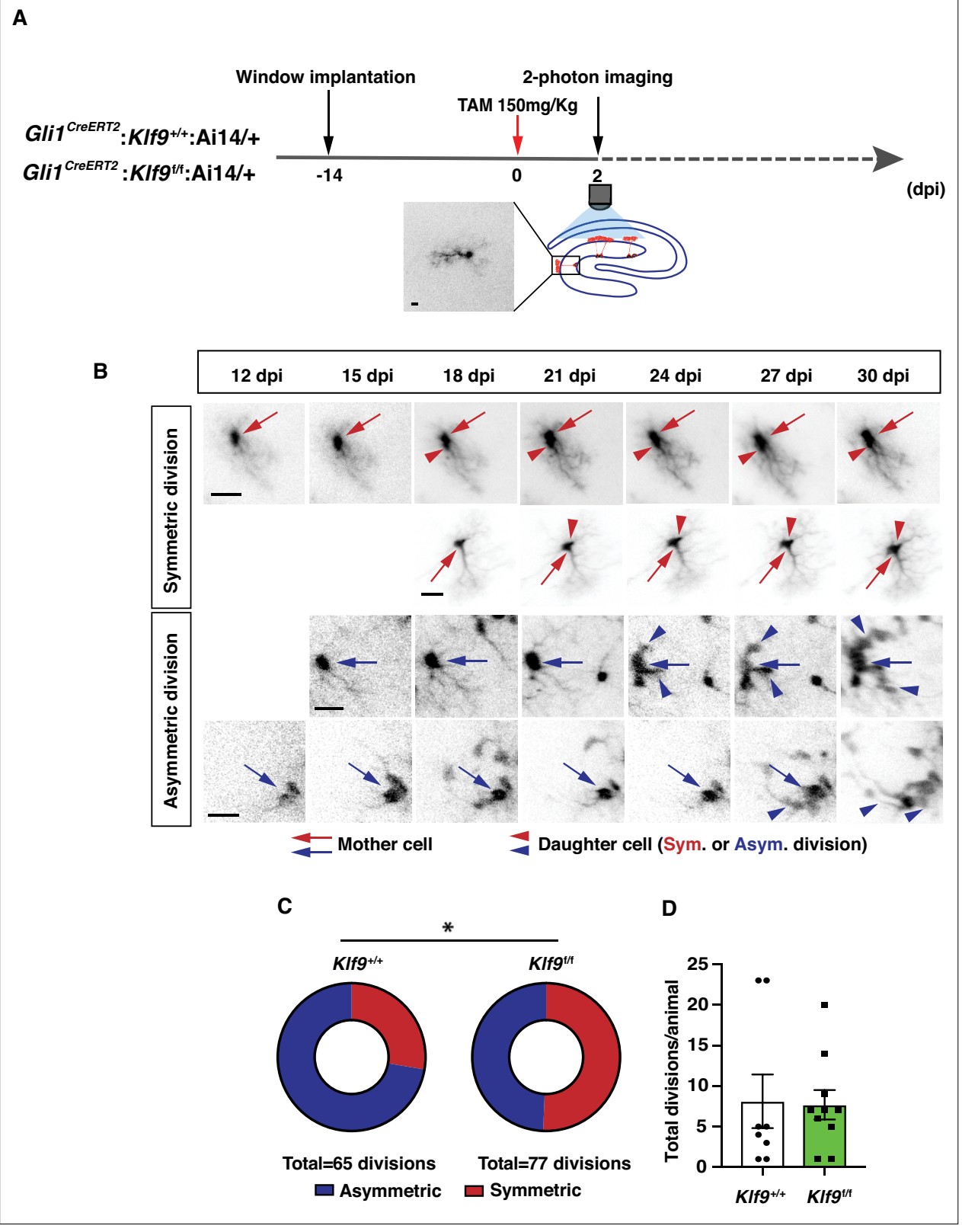

**Figure 3.** Kruppel-like factor 9 (Klf9) functions as a brake on symmetric self-renewal of radial-glial neural stem cells (RGLs). (**A**) Diagram of experimental design for in vivo two-photon imaging experiments. Inset is a high magnification image of a sparsely labeled single RGL in an adult *Gli1^{CreERT2}:Klf9^{+/+}*:Ai14 mouse. (**B**) Representative series of longitudinal imaging from four fields of view showing RGL symmetric and asymmetric divisions. Row 2: control. Rows 1, 3, and 4: experimental. Arrows point to mother cell and arrowheads point to daughter cells. Scale bar: 20 µm. (**C**) Quantification of RGL symmetric

*Figure 3 continued on next page*

*Figure 3 continued*

and asymmetric divisions showing an increase in symmetric divisions in *Gli1$^{CreERT2}$:Klf9$^{f/f}$*:Ai14 mice. *n* = 8 *Gli1$^{CreERT2}$:Klf9$^{+/+}$*:Ai14 mice, 65 divisions; *n* = 10 *Gli1$^{CreERT2}$:Klf9$^{f/f}$*:Ai14 mice, 77 divisions. Odds of symmetric division are 2.7× higher in *Gli1$^{CreERT2}$:Klf9$^{f/f}$*:Ai14 mice, p = 0.015 likelihood-ratio test., *(**D**) Similar number of divisions was recorded for each group to avoid biased assessment of division mode (*n* = 8 and 10 mice/group).

The online version of this article includes the following video and figure supplement(s) for figure 3:

**Figure supplement 1.** Representative images of radial-glial neural stem cell (RGL) divisions captured using two-photon imaging in vivo.

**Figure 3—video 1.** In vivo two-photon imaging of Gli1-postive radial-glial neural stem cells (RGLs).

https://elifesciences.org/articles/72195/figures#fig3video1

**Figure 3—video 2.** In vivo two-photon imaging of Gli1-postive radial-glial neural stem cells (RGLs).

https://elifesciences.org/articles/72195/figures#fig3video2

**Figure 3—video 3.** In vivo two-photon imaging of Gli1-postive radial-glial neural stem cells (RGLs).

https://elifesciences.org/articles/72195/figures#fig3video3

**Figure 3—video 4.** In vivo two-photon imaging of Gli1-postive radial-glial neural stem cells (RGLs).

https://elifesciences.org/articles/72195/figures#fig3video4

their processes. Imaging sessions started 2 days post-tamoxifen injection (dpi) and were repeated daily up to 6 dpi in order to locate isolated labeled RGLs which were clearly identifiable by their tufted radial process. Astrocytes were occasionally labeled but were readily distinguishable from RGLs due to their lack of polar morphology and were disregarded. Post hoc histology analysis of morphological features and immunoreactivity for GFAP in brain sections was performed to corroborate our initial in vivo identification of a subset of RGLs (*Figure 3*, *Figure 3—figure supplement 1*). After 6 dpi we track individual RGLs to quantify their first division event: we revisited each previously identified, RGL containing field-of-view every 3 days and compared it with previous time points in order to quantify the first cell division and classify it as symmetric or asymmetric. As previously described (*Pilz et al., 2018*), asymmetric divisions resulted in motile daughter cells that migrated away from their progenitors within one to two imaging sessions (3–6 days) and exhibited shorter and less stable processes, often undergoing further divisions and differentiation (*Figure 3B*; *Figure 3—video 1*). Conversely, and as shown previously (*Pilz et al., 2018*), symmetric divisions resulted in the appearance of a faint radial process of a single static daughter cell that remained adjacent to its mother cell (*Figure 3B*; *Figure 3—videos 3; 4*). Over time the cell body of the daughter RGL emerged. For our analysis of cell division, we only considered the first division event from an identified RGL, disregarding subsequent divisions of the daughter cells and analysis of RGL-derived lineage trees. Deletion of *Klf9* in RGLs resulted in a significantly greater number of symmetric cell divisions (39 symmetric, 38 asymmetric, 10 mice) compared to *Klf9$^{+/+}$* RGLs (18 symmetric, 47 asymmetric, 8 mice) (*Figure 3C*). We made sure to have a similar number of division events across both genotypes so that we were confident that the differences in the mode of division are not due to under/over sampling each experimental group (N = 8 control mice, 65 divisions, mean 8.125 divisions per mouse; 10 experimental mice, 77 divisions, mean 7.7 divisions per mouse) (*Figure 3D*). These data provide definitive evidence for *Klf9* functioning as a brake on symmetric self-renewal of RGLs in the adult hippocampus.

## *Klf9* regulates a genetic program of RGL activation and expansion

To understand how Klf9 regulates RGL division mode, we performed in vivo molecular profiling of RGLs lacking *Klf9*. We generated *Gli1$^{CreERT2}$:Rpl22HA$^{f/+}$:Klf9$^{f/f\ or\ +/+}$* (B6N.129-*Rpl22tm1.1Psam*/J mice Ribotag) mice (*Sanz et al., 2009*) to genetically restrict expression of a hemagglutinin (HA) epitope-tagged ribosomal subunit exclusively in Gli1+ RGLs (*Figure 4A*). Four days following TAM injections to induce HA expression and Klf9 recombination in a sufficient number of Gli1+ RGLs and progeny arising from first division, we dissected the dentate gyrus subregion, biochemically isolated actively translated transcripts, generated cDNA libraries and performed Illumina sequencing (*Figure 4A–C*). Analysis of the resulting data and gene ontology annotation (gGOSt, https://biit.cs.ut.ee/gprofiler/gost) of differentially expressed genes (DEGs) (*Supplementary file 1*) broadly categorized signaling pathways and molecular programs associated with NSC activation and quiescence (*Mukherjee et al., 2016*; *Zhang et al., 2019*; *Mira et al., 2010*; *Codega et al., 2014*; *Shin et al., 2015*; *Hochgerner et al., 2018*; *Knobloch et al., 2013*; *Figure 4C*; *Figure 4—figure supplement 1*, *Supplementary files 2 and 3*). Functional categories enriched among upregulated DEGs (276) included phospholipase activity

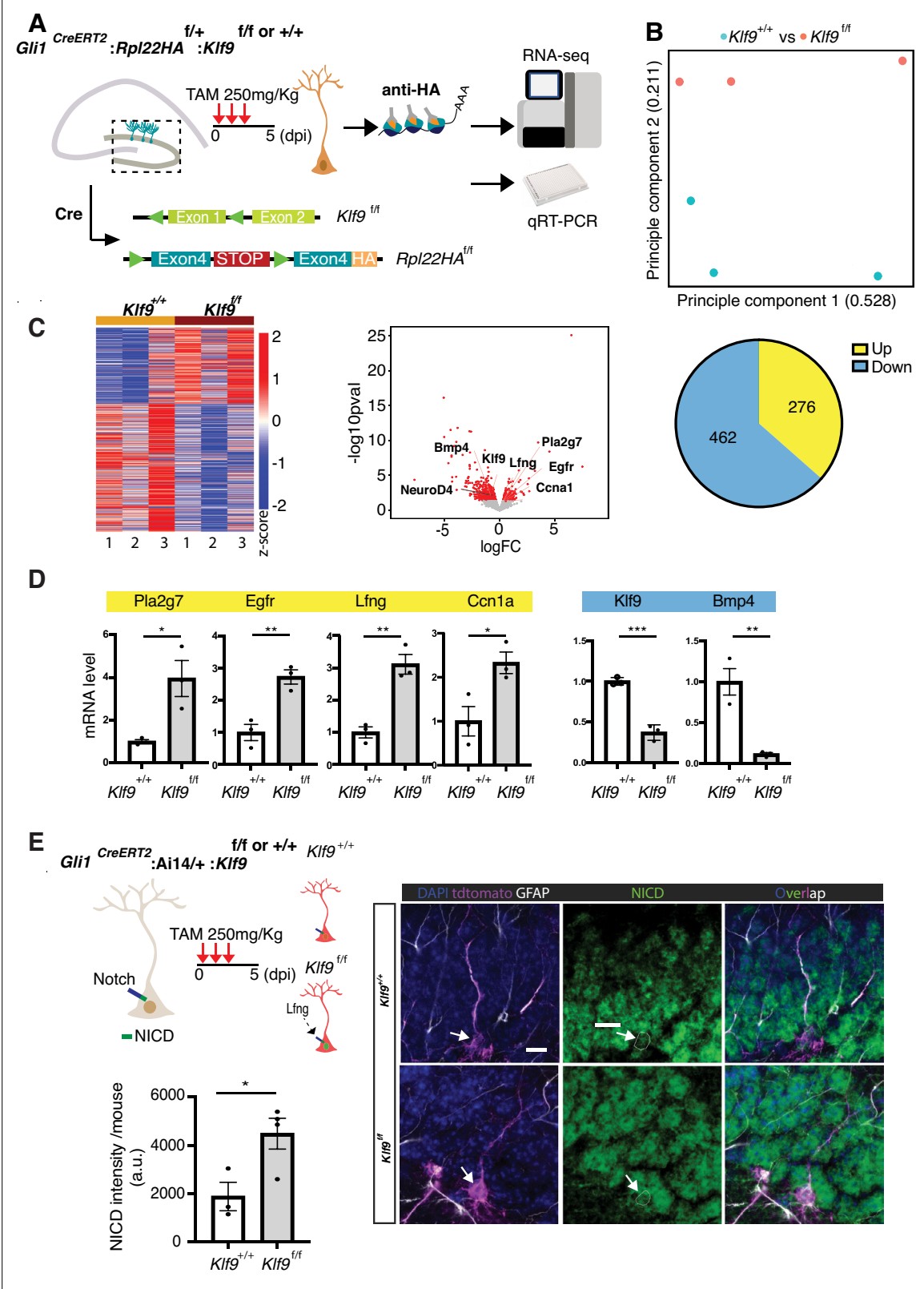

**Figure 4.** Kruppel-like factor 9 (*Klf9*) regulates genetic programs underlying radial-glial neural stem cell (RGL) expansion. (**A**) Schematic of experimental workflow to biochemically isolate and sequence translated mRNAs from Gli1+ RGLs (*Gli1^CreERT2^:Rpl22HA^f/+^:Klf9^f/f or +/+^* mice). *n* = 3 mice, 6 dentate gyri/sample, 3 samples per group. (**B**) Principal component analysis (PCA) plot of translational profiles of *Gli1^CreERT2^*-targeted *Klf9^+/+ or f/f^* RGLs. First two principal components are shown with the corresponding fractions of variance. (**C**) Left: heatmap of expression values for differentially expressed

*Figure 4 continued on next page*

*Figure 4 continued*

genes. Middle: volcano plot of statistical significance (−log10 p value) vs. magnitude of change (log2 of fold change) of gene expression. Differentially expressed genes are marked in red. Upregulated genes in *Klf9*^f/f RGLs are on the right and downregulated genes in *Klf9*^f/f RGLs are on the left. Right: pie chart of numbers of upregulated and downregulated genes in *Gli1*^CreERT2-targeted *Klf9*^f/f RGLs. (**D**) qRT-PCR on biochemically isolated mRNAs from *Gli1*^CreERT2:*Rpl22HA*^f/+:*Klf9*^f/f or +/+ mice validating candidate differentially expressed genes. n = 3 samples, 6 dentate gyri/sample, 3 samples per group. (**E**) Immunostaining and quantification of Notch1 intracellular domain (NICD) in RGLs of *Gli1*^CreERT2: *Klf9*^f/f or +/+ mice. Deletion of *Klf9* results in increased NICD levels in RGLs consistent with Lnfg-dependent potentiation of Notch1 signaling (Cartoon, top left). n = 3 mice/group. Data are represented as mean ± standard error of the mean (SEM). *p < 0.05, **p < 0.01, ***p < 0.001. Scale bar: 10 μm.

The online version of this article includes the following figure supplement(s) for figure 4:

**Figure supplement 1.** Annotation of upregulated and downregulated differentially expressed genes (DEGs) in Gli1+ radial-glial neural stem cells (RGLs) following Kruppel-like factor 9 (*Klf9*) deletion.

(Pla2g7, Pla2g4e, and Gpld1), mitogen growth factor signaling (Egfr, Fgfr3, Ntrk2, and Lfng), and ligand-gated ion channels (Gabra1, Chrna7, Grin2C, and P2$r$ × 7). Additionally, our analysis revealed elevation of metabolic programs sustaining energy production and lipogenesis through generation of Acetyl-CoA: CoA- and fatty acid-ligase activity (Acsl3, Ascl6, Acss1, and Acsbg1) and oxidoreductase and aldehyde dehydrogenase activity (Acad12, Acox1, Ak1b10, Aldh3b1, and Aldh4a1) (*Knobloch et al., 2013*; *Namba et al., 2021*; *Knobloch et al., 2017*; *Xie et al., 2016*; *Supplementary file 2*). A complementary set of modules overrepresented in the downregulated gene set (462 DEGs) were quiescence growth factor signaling (Bmp2 and Bmp4), extracellular matrix binding (Itga3, Itga10, and Igfbp3, 6, 7), cell adhesion (e.g., Emb, Itga3, and Itga5), actin binding (Iqgap1), TFs (NeuroD4 and Zic3), and voltage-gate potassium channel activity (Kcnj8 and Kcnq1) (*Supplementary file 3*).

For validation of DEGs previously linked with NSC quiescence and activation (*Mukherjee et al., 2016*; *Zhang et al., 2019*; *Codega et al., 2014*; *Shin et al., 2015*; *Knobloch et al., 2013*; *Baser et al., 2019*), we performed qRT-PCR on an independent replicate of biochemically isolated mRNAs from this population of Gli1+ RGLs in vivo. We first confirmed downregulation of Klf9 in RGLs. Next, we validated downregulation of canonical quiescence signaling factors (Bmp4) and upregulation of genes involved in lipid metabolism (Pla2g7), cell cycle (Ccn1a), mitogen signaling (epidermal growth factor receptor, Egfr), and Notch signaling (Lunatic fringe, Lfng) (*Figure 4D*). Consistent with Lfng-mediated potentiation of Notch1 signaling through cleavage of the Notch1 intracellular domain (NICD), we observed significantly elevated levels of NICD in Gli1+ RGLs lacking Klf9 (*Figure 4E*; *Hochgerner et al., 2018*; *Zhao and Wu, 2018*). We infer from our loss-of-function data that high levels of Klf9 in RGLs induce BMP4 expression and repress gene modules specifying mitogen signaling, fatty acid oxidation, RGL differentiation, and cell-cycle exit to inhibit RGL expansion.

## Discussion

Central to experience-dependent regulation of neurogenesis is the ability of RGLs to constantly balance demands for neurogenesis and astrogenesis or RGL expansion with self-preservation through regulation of quiescence. Since interpretation of the external world is dependent on integration and convergence of physiological extracellular signals upon TFs in RGLs, enriching and adverse experiences are likely to modulate the balance between transcriptional programs that regulate RGL division modes supporting amplification or asymmetric self-renewal (*Vicidomini et al., 2020*). However, in contrast to our knowledge of TFs that regulate asymmetrical self-renewal of RGLs in the adult hippocampus (*Mukherjee et al., 2016*; *Jones et al., 2015*; *Zhang et al., 2019*; *Ehm et al., 2010*; *Imayoshi et al., 2010*), the identities of transcriptional regulators of symmetric self-renewal of RGLs have remained elusive. By combining conditional mouse genetics with in vivo clonal analysis and longitudinal two-photon imaging of RGLs, we demonstrated that Klf9 acts as a transcriptional brake on RGL activation state and expansion through inhibition of symmetric self-renewal (*Figure 5*).

That Klf9 expression is higher in nondividing RGLs than in activated RGLs is consistent with gene expression profiling of quiescent adult hippocampal RGLs (*Bottes et al., 2021*; *Knobloch et al., 2013*; Jaeger and Jessberger, personal communication) and other quiescent somatic stem cells such as satellite cells (*Pallafacchina et al., 2010*) and NSCs in the subventricular zone (*Codega et al., 2014*; *Morizur et al., 2018*; *Renault et al., 2009*). Loss of Klf9 in Gli1+ RGLs resulted in increased RGL activation. Based on our clonal analysis of RGL output and in vivo translational profiling, we think that this

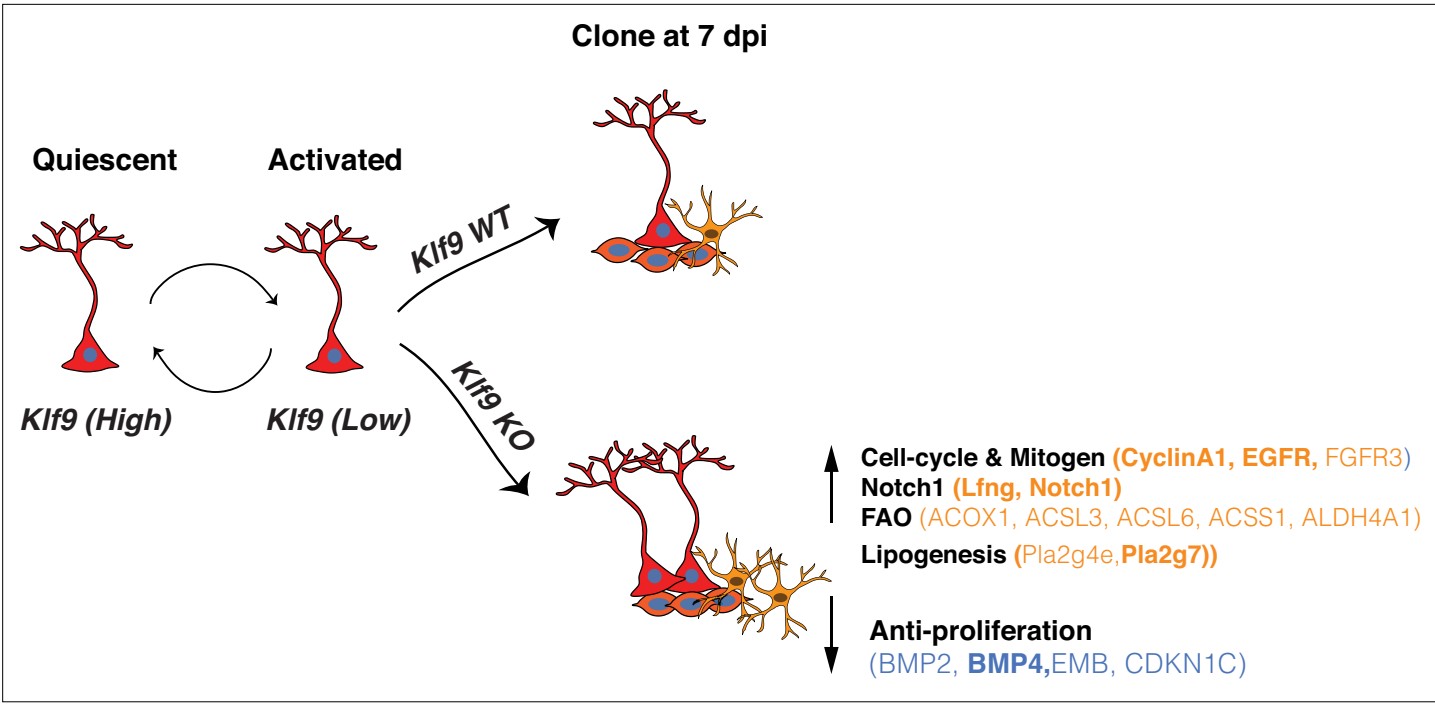

**Figure 5.** Summary schematic conveying Kruppel-like factor 9 (*Klf9*) functions in radial-glial neural stem cell (RGL) activation and self-renewal RGLs integrate extracellular, experiential signals to exit quiescence, the dominant state, and become activated. *Klf9* expression is elevated in quiescent RGLs. Low levels of *Klf9* in RGLs are associated with increased activation. Once activated, RGLs lacking *Klf9* are biased toward symmetric self-renewal and RGL expansion. Translational profiling of RGLs reveals how loss of *Klf9* results in downregulation of a program of quiescence and activation of genetic (mitogen, notch) and metabolic (fatty acid oxidation and lipid signaling) programs underlying RGL symmetric self-renewal. Candidate differentially expressed upregulated (orange) and downregulated genes (blue) in RGLs following *Klf9* deletion are shown here. Genes in bold indicate validation by qRT-PCR.

increased RGL activation reflects maintenance of an activated or cycling state (also discussed later) to support increased symmetric self-renewal (*Encinas et al., 2011*).

Our current knowledge of TFs that regulate symmetric self-renewal in the adult hippocampus can only be extrapolated from studies on hippocampal development (*Noguchi et al., 2019*). Clonal analysis of Gli1-targeted RGLs revealed multi-RGL containing clones with progeny. This potentially reflects competition between TFs that dictate balance between symmetric and asymmetric divisions, compensation by downstream effectors of Klf9 or constraints on RGL expansion imposed by availability of niche factors. Such compensatory mechanisms may also explain why constitutive deletion of *Klf9* does not overtly affect size of the dentate gyrus (*Scobie et al., 2009*).

Studies on adult hippocampal neural stem and progenitor cells have relied on assays that induce quiescence and activation in vitro (*Knobloch et al., 2013*), unbiased single cell profiling of neurogenesis (*Shin et al., 2015*; *Hochgerner et al., 2018*) or FACS sorting of neural stem and progenitor cells in vivo (*Zhang et al., 2019*). Because asymmetric self-renewal is the predominant mode of division, it is most certainly the case that the RGL activation profile inferred from these studies is biased toward asymmetric, rather than symmetric, self-renewal. In contrast, our in vivo translational profiling of long-term self-renewing Gli1+ RGL population following cell-autonomous deletion of Klf9 allowed us to infer how changes in gene expression relate to RGL symmetric division mode and create an exploratory resource for the NSC research community. While ribosomal profiling does not allow us to isolate transcripts from single RGLs, it offers other advantages such as minimizing stress response associated with cell dissociation (*Machado et al., 2021*). Since *Gli1Cre^{ERT2}* specifically targets RGLs and astrocytes (but not progenitors) and we performed biochemical profiling at 4 days postrecombination when we first observe RGL derived progeny, our analysis largely reflects changes in the RGL population, progeny arising from first division and astrocytes. That we observe an enrichment of genes expressed exclusively in RGLs permits us to link gene expression with changes in RGL numbers driven by division mode. Analysis of Klf9 levels by qRT-PCR suggests greater than 50% recombination

efficiency of Klf9 in targeted cell populations. Our genome-wide expression analysis suggests that Klf9 functions as an activator or repressor depending on cellular context, although repression appears to be the dominant mode of gene regulation (*Knoedler et al., 2017*; *Ying et al., 2014*). Validation of specific DEGs in biochemically isolated transcripts from RGLs suggests that Klf9 may activate BMP4 expression in RGLs to suppress activation in vivo (*Mira et al., 2010*). Additionally, Klf9 suppresses RGL proliferation through repression of mitogen signaling receptor tyrosine kinases (EGFR), lipidogenesis (Pla2g7), and cell cycle (CyclinA1). Pla2g7, interestingly, is expressed only in RGLs and astrocytes in the DG (*Shin et al., 2015*; *Hochgerner et al., 2018*) and as such may represent a novel marker of activated RGLs. Given the dual roles of Notch signaling in regulation of active and quiescent RGLs (*Sueda and Kageyama, 2020*), we validated that Lfng is significantly upregulated in RGLs lacking Klf9. Lfng is exclusively expressed in RGLs, promotes Notch1 signaling through glycosylation of Notch1 and generation of NICD following ligand binding, and dictates RGL activation in a ligand-dependent manner (*Semerci et al., 2017*). Genetic overexpression of Lfng in T-cell progenitors sustained Notch1-mediated self-renewal and clonal expansion at expense of differentiation (*Yuan et al., 2011*). Consistent with Lfng upregulation in RGLs, we observed elevated levels of NICD in Gli1+ RGLs lacking Klf9 indicative of enhanced Notch1 signaling.

Bioinformatics analysis of our data identified enhanced fatty acid β-oxidation (FAO), a substrate for energy production and lipogenesis as a metabolic program recruited to sustain RGL expansion (*Figure 5*). In fact, lineage tracing studies on embryonic neocortical NSCs have demonstrated a role for FAO in maintenance of NSC identity and proliferation (*Namba et al., 2021*). Specifically, inhibition of Tmlhe (a carnitine biosynthesis enzyme) and carnitine-dependent long-chain FAO (carnitine palmitoyltransferase I, CPT1, which catalyzes the rate-limiting reaction in this process) resulted in a marked increase in symmetric differentiating divisions at expense of both symmetric and asymmetric self-renewal of NSCs (*Xie et al., 2016*). Inhibition of FAO prevented hematopoietic stem cell maintenance and promoted symmetric differentiating divisions of hematopoietic stem cells (*Ito et al., 2012*). High levels of FAO are directly linked to intestinal stemness (*Mana et al., 2021*) and persistence of proliferative capacity across cancers (*Oren et al., 2021*). In sharp contrast to these findings, it has been suggested that high levels of FAO are important for maintaining RGL quiescence. Specifically, deletion of Cpt1a (and inhibition of FAO) in adult hippocampal NSC and progenitors impaired expansion and reduced numbers of RGLs. However, it could not be determined if this was due to death and/or inhibition of symmetric self-renewal of RGLs (*Knobloch et al., 2017*). Based on our data, we propose that NSCs, like other somatic stem cells and progenitors, require high levels of FAO for symmetric self-renewal or expansion.

How does Klf9 function as a brake on RGL symmetric self-renewal? We propose that Klf9 corepresses a suite of genes associated with maintenance of RGLs in symmetric division mode. Pioneering studies have implicated Notch signaling in sustaining symmetric divisions of neuroepithelial cells (*Egger et al., 2010*), expansion of putative NSCs and progenitors (*Androutsellis-Theotokis et al., 2006*) and maintenance of radial glial cell like identity through inhibition of differentiation and cell-cycle exit (*Gaiano et al., 2000*; *Yoon et al., 2008*). Importantly, genetic gain-of-function of Notch1 signaling in RGLs in the adult DG maintains RGLs at the expense of hippocampal neurogenesis (*Breunig et al., 2007*). Klf9 may also directly suppress a proneurogenic program in RGLs (e.g., NeuroD4, downregulated DEG, *Supplementary file 3*; *Masserdotti et al., 2015*) or indirectly via competitive interactions with TFs that regulate RGL asymmetric self-renewal. Taken together, loss of Klf9 in RGLs drives expansion through enhanced mitogen and cell-cycle signaling (*Berdugo-Vega et al., 2020*), prevention of RGL differentiation, and elevation of lipogenic and FAO metabolic programs (*Figure 5*).

Our findings stimulate discussion on how experiential signals regulate RGL activation and expansion. To date, GABA(A) R signaling and PTEN signaling (by inhibiting PI3K–Akt pathway) have been shown to promote quiescence and suppress RGL amplifying divisions (*Bonaguidi et al., 2011*; *Song et al., 2012*). It is plausible that Klf9 participates in these signaling pathways as a downstream actuator. *Klf9* expression is reduced in NSCs lacking FoxO3 (*Renault et al., 2009*). Thus, Akt-dependent regulation of NSC activation through inactivation of FoxO3 (*Urbán et al., 2019*) may require *Klf9* downregulation. Since some of the identified Klf9 target genes are also regulated by other TFs (e.g., inhibition of EGFR and cyclinA1 by Notch2 [*Zhang et al., 2019*], activation of Pla2g7 by FoxO3 [*Renault et al., 2009*]), we infer that these factors do not compensate each other, but instead, confer flexible integration of diverse physiological signals in RGLs to regulate activation. Inhibition of pulsatile

glucocorticoid receptor signaling has also been shown to promote RGL quiescence (*Schouten et al., 2020*). Because Klf9 expression is regulated by steroid hormone signaling and neural activity (*Scobie et al., 2009*; *Datson et al., 2011*; *Besnard et al., 2018*) and Klf9 represses gene expression through recruitment of a mSin3A corepressor complex (*Zhang et al., 2001*), Klf9 may support an epigenetic mechanism for reversible, experiential regulation of NSC decision making.

Our genome-wide dataset serves as a general exploratory community resource in several ways. First, it catalyzes further enquiry into mechanisms underlying NSC quiescence and expansion. By way of example, candidate genes such as the cell adhesion molecule Embigin (downregulated DEG) regulates quiescence of hematopoietic stem/progenitor cells (*Silberstein et al., 2016*) whereas the alpha7 nicotinic receptor (upregulated DEG), ChrnA7, has been shown to be required for maintaining RGL numbers (*Otto and Yakel, 2019*). Second, numerous genes identified in our blueprint are implicated in driving tumorigenesis and as such may guide differentiation-based strategies to block tumor proliferation (*Carracedo et al., 2013*). Third, our work motivates assessment of how Klf9 may link extracellular, physiological signals with genetic and metabolic programs in RGLs. Fourth, our findings may guide investigation of functional significance of Klf9 enrichment in other quiescent neural (SVZ) (*Codega et al., 2014*; *Morizur et al., 2018*; *Renault et al., 2009*) and somatic stem cell populations (*Pallafacchina et al., 2010*).

Our study enables a more holistic assessment of how competing transcriptional programs in RGLs mediate decision making by including regulators of symmetric and asymmetric self-renewal. A deeper understanding of Klf9-dependent regulation of RGL homeostasis may guide genetic and metabolic strategies to replenish the RGL reservoir and restore neurogenesis following injury or expand the NSC pool in anticipation of future neurogenic demands to support hippocampal-dependent memory processing and emotional regulation (*Anacker and Hen, 2017*; *Miller and Sahay, 2019*; *McAvoy et al., 2016*).

## Materials and methods

Animals were handled and experiments were conducted in accordance with procedures approved by the Institutional Animal Care and Use Committee at the Massachusetts General Hospital and Albert Einstein College of Medicine in accordance with NIH guidelines. Mice were housed three to four per cage in a 12 hr (7:00 a.m. to 7:00 p.m.) light/dark colony room at 22–24°C with ad libitum access to food and water.

### Mouse lines

The following mouse lines were obtained from Jackson Labs: *Klf9-lacZ* knock-in (Stock No. 012909), *Gli1$^{CreERT2}$* (Stock No. 007913), Ai14 (Stock No. 007908), mT/mG (Stock No. 007676), B6N.129-*Rpl22$^{tm1.1Psam}$*/J (RiboTag) (Stock No. 011029), and *POMC*-Cre (Stock No. 010714). *Sox1tTA* transgenic mice (*Venere et al., 2012*) were obtained from Dr. Robert Blelloch (University of California, San Fransisco). *Klf9$^{LacZ/LacZ}$* mice were obtained from Dr. Yoshiaki Fujii-Kuriyama (University of Tsukuba and is also available from Jackson Labs, Stock No. 012909). tetO Kf9/Klf9 knockin mice were generated by us previously (*McAvoy et al., 2016*). Nestin GFP mice (*Mignone et al., 2004*) were obtained from Dr. David Scadden at MGH.

*Klf9* conditional knockout mice were generated through homologous gene targeting using C57BL/6 ES cells by Cyagen. F0s were bred with C57BL/6J mice to generate F1s with germline transmission and mice were backcrossed with C57BL/6J mice for 5+ generations. A set of primers (forward: GGTAGTCAAATGGCGCAGCTTTT; reverse: CCATCCATTCCTTCATCAGTCTCC) was used to genotype *Klf9$^{+/+ or f/f}$* mice to amplify 363 bp mutant band and 240 bp wildtype band. *Gli1$^{CreERT2}$:Klf9$^{+/+ or f/f}$* Ai14 and *Gli1$^{CreERT2}$:Klf9$^{+/+ or f/f}$*:mT/mG+/−, were generated by crossing *Gli1$^{CreERT2}$* mice with mT/mG or Ai14 and *Klf9$^{+/+ or f/f}$* mice in a C57BL/6J background.

### BrdU administration

For analysis of cell proliferation in dentate gyrus, mice were injected with BrdU (200 mg/kg body weight, i.p.) and sampled 2 hr later. For analysis of long-term retaining cells in dentate gyrus, mice were given daily injection of BrdU (25 mg/kg body weight, i.p.) for 14 days and sampled 24 hr after the last injection.

## Tamoxifen administration

Tamoxifen (20 mg/ml, Sigma, T5648) was freshly prepared in a 10% ethanol of corn oil (Sigma C8267). For population analysis, a dose of 150 or 250 mg/kg was intraperitoneally injected into 8-week-old male and female mice (*Figure 1F*). For clonal analysis, a dose of 50 and 100 mg/kg were used in reporter lines of Ai14 and mT/mG, respectively (*Figure 2A, E*). Mice were sampled 7 or 28 days post-tamoxifen injection. For two-photon imaging (*Figure 3A*), one dose of 150 mg/kg tamoxifen was given 2 days prior to in vivo imaging. For ribosomal profiling, a dose of 250 mg/kg body weight was intraperitoneally injected into 2–3 months mice every 12 hr for three times. Mice were sampled 4 days after the last injection (*Figure 4A*).

## Tissue processing and immunostaining

35 µm cryosections obtained from perfused tissue were stored in phosphate-buffered saline (PBS) with 0.01% sodium azide at 4°C. For immunostaining, floating sections were washed in PBS, blocked in PBS containing 0.3% Triton X-100% and 10% normal donkey serum and incubated with primary antibody overnight at 4°C overnight (Rockland, rabbit anti RFP, 1:500; Millipore, chicken anti-GFAP, 1:2000; goat anti-GFP, Novus, 1:500; Santa Cruz, sc-8066, Goat anti-DCX, 1:500). The Mcm2 (BD Biosciences, mouse anti-Mcm2; 1:500), GFP (Abcam, Chicken anti-GFP, 1:2000), LacZ (Promega, Mouse anti-beta Galactosidase, 1:2000), and Nestin (Aves lab, chicken anti-Nestin, 1:400) antigens were retrieved by incubating brain sections in citric buffer in pressure cooker (Aprum, 2100 retriever) for 20 min, followed by 60 min cooling to room temperature. BrdU antigen was retrieved by incubating brain sections in 2 N HCl for 30 min at 37°C following 15 min fixation in 4% paraformaldehyde (PFA on previously processed fluorescent signal). On the next day, sections were rinsed three times for 10 min in PBS and incubated for 90 min with fluorescent-label-coupled secondary antibody (Jackson ImmunoResearch, 1:500). Sections were rinsed three times for 10 min each in PBS before mounting onto glass slides (if applicable) and coverslipped with mounting media containing DAPI (Fluoromount with DAPI, Southern Biotech). NICD (rabbit anti-cleaved Notch1, Assay Biotech Cat# L0119 RRID:AB_10687460 at 1:100) immunostaining was performed as described (*Semerci et al., 2017*).

## *Klf9* ISH

We used a transgenic mouse line that expresses GFP under the control of the Nestin promoter to label the cell bodies (*Mignone et al., 2004*). Mice were sacrificed 2 hr after a single BrdU injection (200 mg/kg). *Klf9* expression was detected by florescent in situ hybridization (FISH) using a *Klf9* antisense probe complementary to exon 1 (530–1035 bp) of *Klf9* mRNA. Briefly, ISH was performed using dioxygenin-labeled riboprobes on 35 µm cryosections generated from perfused tissue as described (*McAvoy et al., 2016*). Premixed RNA labeling nucleotide mixes containing digoxigenin-labeled UTP (Roche Molecular Biochemicals) were used to generate RNA riboprobes. *Klf9* null mice were used as a negative control and to validate riboprobe specificity. Riboprobes were purified on G-50 Microspin columns (GE Healthcare). Probe concentration was confirmed by Nanodrop prior to the addition of formamide. Sections were mounted on charged glass (Superfrost Plus) slides and postfixed for in 4% PFA. Sections were then washed in DEPC-treated PBS, treated with proteinase K (40 µg/ml final), washed again in DEPC-treated PBS, and then acetylated. Following prehybridization, sections were incubated with riboprobe overnight at 60°C, washed in decreasing concentrations of SSC buffer, and immunological detection was carried out with anti-DIG peroxidase antibody (Roche) at 4°C overnight and were visualized using Cy3-conjugated Tyramide Signal Amplification system (Perkin-Elmer) at room temperature. ISH was followed by immunostaining for GFP (Goat anti-GFP, Novus, 1:500) and BrdU (Rat anti-BrdU, Biorad, 1:500) incubated at 4°C overnight and followed by incubation of 488- and Cy5-conjugated secondary antibodies (Jackson ImmunoResearch, 1:500) for 2 hr at room temperature. *Klf9* ISH was performed on *POMC*-Cre:*Klf9*$^{+/+ \text{ and } f/f}$ mice using *Klf9* exon1 probe to validate the Klf9 conditional knockout mice. Immunological detection was carried out with anti-DIG antibody conjugated with alkaline phosphatase (Roche). Color reaction was conducted with NBT/BCIP. *Klf9* null mice were used as a negative control.

Estimation of *Klf9* recombination frequency in Gli1-positive tdTomato-labeled RGLs. *Gli1*$^{CreERT2}$:*Klf9*$^{+/+}$:Ai14 and *Gli1*$^{CreERT2}$:*Klf9*$^{f/f}$:Ai14 mice were given one dose of tamoxifen (150 mg/kg) IP and were then perfused with DPEC-treated PBS and fixed with 4% PFA. 35 µm cryosections sections were mounted on the same slides. After hybridization with *Klf9* riboprobe, slides were washed and

blocked with NEN buffer for 1 hr at RT. The following antibodies were used for immunostaining: anti-DIG peroxidase antibody (mouse, 1/8000, Roche); anti-RFP (rabbit, 1/500, Rockland). Slides were washed and coverslipped with mounting medium (Southern biotech). *Klf9* fluorescence intensity within the cell body was recorded.

### Images acquisition and analysis

Images were obtained from one set of brain sections (six sets generated for each brain) for each immunostaining experiment (set of antigens). Stained sections were imaged at ×20 or ×40 on a Nikon A1R Si confocal laser, a TiE inverted research microscope or a Leica SP8 confocal microscope. All of analysis were performed by an experimenter blind to group identity.

LacZ intensity quantification. We used mice carrying a LacZ allele knocked into the endogenous *Klf9* allele (*Klf9*$^{LacZ/LacZ\ or\ LacZ/+}$ mice) (*Scobie et al., 2009*). *Klf9*$^{Lac/LacZ\ or\ LacZ/+}$ mice were crossed with Nestin GFP mice to generate *Klf9*$^{LacZ//+}$;Nestin GFP mice. These mice were used to quantify *LacZ* expression levels in quiescent RGLs (GFP+ MCM2 with radial process), activated RGLs (GFP+ MCM2+ with radial process), and neural progenitor cells (NPCs; (GFP+ MCM2+ without radial processes)). The distinction between RGLs and NPCs was determined through morphological analysis. Images (1024 resolution) were acquired as 7 Z-stacks with a step size of 1 μm. Two to four stacks of images from each mouse were selected for further quantification. Since the *LacZ* gene had been knocked into the endogenous *Klf9* locus, mean intensity of *Lac*Z expression, assessed by fluorescent signal with LacZ immunostaining using ImageJ software in each GFP+ cell body, was used as a surrogate for *Klf9* expression in *Klf9*$^{LacZ/+}$ mice. Mean background intensity was obtained from *LacZ* negative regions being divided from the calculations in the same section.

*Klf9* FISH signal quantification. Images (2048 resolution) were acquired by a Leica SP8 confocal microscope as 30 Z-stacks with a step size of 0.5 μm. Representative images were generated by exporting stacked confocal images at full resolution for three-dimensional visualization using Imaris. The distinction between NPCs and NSCs was determined through morphological analysis with GFP staining. Activated RGLs were differentiated from quiescent RGLs through BrdU antibody staining (cell proliferation markers). Analysis and quantification of *Klf9* signal intensity in each GFP+ cell body were conducted using automatic counting with ImageJ software. Images were converted into 1-bit images. Then *Klf9* puncta were counted within GFP+ cell body boundaries through particle analysis allowing for number and average size of puncta to be recorded. Klf9 null mice crossed with Nestin GFP mice were used as a negative control.

### Clonal lineage analysis

Clonal analysis was conducted with sparse labeling after optimizing dose of tamoxifen as previously described (*Bonaguidi et al., 2011*). Ai14 and mTmG reporter mice were used to visualize the recombined cells. Serial coronal sections were generated and immunostained for GFAP, RFP, or GFP antigens. Images acquisition and analysis were restricted to entire dentate gyri ~2000 μm along the dorsal–ventral axis. RGLs were classified as cells that were located in the subgranular zone, had radial projections that extended into the granule cell layer, and were colabeled with GFAP and RFP or GFP. Cells with GFAP labeling without radial processes but exhibiting a bushy morphology were identified as astrocytes. Recombined GFP+ or RFP+ cells without GFAP labeling in close spatial proximity to other cells were identified as neuronal progeny cells. A ring with a radius of 50 μm from the center of the RGL was used to determine the clone composition. A single cell (astrocyte or neuron) was not counted as a clone. Images (1024 resolution) were acquired using a Leica SP8 confocal microscope as 20–25 Z-stacks with a step size of 1.5 μm. Mice with less than two clones per hemisection on average were determined as standard for sparse labeling and were selected for clonal analysis. Except for the single RGL clone category, all the labeled cells within one clone were in close spatial proximity to each other. Clones were categorized according to the presence or absence of an RGL and the type of progeny. For imaris image analysis, Z-series confocal images were processed for all the channels. The intensity of each channel was adjusted and representative images were used to generate a TIFF file by taking a 'screen snapshot'.

### Two-photon imaging of Gli1+ *Klf9*+/+ or f/f RGLs division modes in vivo

Twelve- to sixteen-week-old *Gli1*$^{CreERT2}$:*Klf9*$^{+/+\ or\ f/f}$:Ai14 mice were used for intravital 2P imaging of RGLs.

Window implantation: We followed an established protocol to implant a cranial window over the right hemisphere of the dorsal hippocampus (*Pilz et al., 2018*). Briefly, we drilled a ~3-mm wide craniotomy, removed the underlying dura mater and aspirated the cortex and corpus callosum. A 3-mm diameter, 1.3-mm deep titanium implant, with a glass sealed to the bottom was then placed above the hippocampus. The implant and a titanium bar (29 × 3.8 × 1.3 mm) were held in place with dental cement. A titanium bar was used in order to secure the animal to the microscope stage. Mice were given a single dose of dexamethasone (1 mg/kg, i.p.) before surgery to reduce brain swelling, and carprofen (5 mg/kg, i.p.) for inflammation and analgesic relief after surgery completion. Implanted animals were given 2 weeks to recover from surgery and allow any inflammation to subside.

Two-photon imaging of aRGL divisions: In vivo imaging was done on a custom two-photon laser scanning microscope (based on Thorlabs Bergamo) using a femtosecond-pulsed laser (Coherent Fidelity 2, 1075 nm) and a ×16 water immersion objective (0.8 NA, Nikon). We imaged mice under isoflurane anesthesia (~1% isoflurane in $O_2$, vol/vol) and head-fixed to the microscope stage via a titanium bar implant while resting on a 37°C electrical heating pad (RWD ThermoStar). Expression of the tdTomato fluorescent label in Gli1+ RGLs was induced with a single injection of Tamoxifen (150 µl/mg) 2 weeks after window implantation. Imaging began 2 days after tamoxifen injection (2 dpi) and continued every day until 6 dpi in order to locate sparse labeled RGLs. Afterwards, mice were imaged every 3 days, whenever possible and were imaged up to 60 days. Using a coordinate system, we marked locations of RGLs for recurrent imaging of the same cell. At each time point, we acquired a three-dimensional image stack of each field of view containing tdTomato-expressing cells and annotated their location so that the same cell could be imaged again in the following session.

Cell division classification: Cell divisions were analyzed by two different experimenters blinded to genotype. We first compiled all Z-stacks into a single sum-projected image for each time point, and then we used FIJI-ImageJ to analyze the images. Only the first recorded cell division for a given clone was included in the analysis. We defined RGL symmetric division as a new RGL generated from the mother RGL, characterized by the development of a stable radial process and static behavior of cell bodies for at least 6 days after birth. We defined asymmetric division as new NPCs generated from the mother RGL that exhibited shorter and less stable processes. These NPCs often began to migrate away within one to two imaging sessions (3–6 days).

## Ribotag isolation of mRNAs from Gli1+ RGLs

We used *Gli1^{CreERT2}:Rpl22HA^{f/+}:Klf9^{f/f or +/+}* mice which enables expression of HA-tagged ribosomal protein RPL22 (RPL22–HA) following Cre recombination in Gli1+ *Klf9^{f/f or +/+}* RGLs. RiboTag immunoprecipitation and RNA extraction were performed 4 days after last TAM injection following the original protocol with minor modifications (*Sanz et al., 2009*). Six dentate gyri from three mice were pooled per sample and homogenized with a dounce homogenizer in 900 µl cycloheximide-supplemented homogenization buffer. Homogenates were centrifuged and the supernatant incubated on a rotator at 4°C for 4 hr with 9 µl anti-HA antibody (CST Rb anti-HA #3724, 1:100) to bind the HA-tagged ribosomes. Magnetic IgG beads (Thermo Scientific Pierce #88847) were conjugated to the antibody–ribosome complex via overnight incubation on a rotator at 4°C. RNA was isolated by RNeasy Plus Micro kit (Qiagen 74034) following the manufacturer's protocol. Eluted RNA was stored at −80°C. For enrichment analysis, 45 µl of homogenate (pre- anti-HA immunoprecipitation) was set aside after centrifugation, kept at −20°C overnight, and purified via RNeasy Micro kit as an 'input' sample, and used to determine NSC enrichment. RNA quantity and quality were measured with a Tape Station (Agilent) and Qubit fluorimeter (Thermo Fisher Scientific). Sequencing libraries were prepared using Ultra Low Input RNA Kit (Clontech).

## RNA-seq analysis

NGS libraries were constructed from total RNA using Clontech SMARTer v4 kit (Takara), followed by sequencing on an Illumina HiSeq 2500 instrument, resulting in 20–30 million 50 bp reads per sample. The STAR aligner (*Dobin et al., 2013*) was used to map sequencing reads to transcriptome in the mouse mm9 reference genome. Read counts for individual genes were produced using the unstranded count function in HTSeq v.0.6.0 (*Anders et al., 2015*), followed by the estimation of expression values and detection of differentially expressed transcripts using EdgeR (*Robinson et al., 2010*) and including only the genes with count per million reads >1 for one or more samples (*Anders*

*et al., 2013*). DEGs were defined by at least 1.2-fold change with p < 0.05. NCBI GEO accession number GSE164889.

qRT-PCR mRNA was biochemically pooled and isolated as described above for ribosomal profiling. The first-stranded complementary DNA was generated by reverse transcription with SuperScript IV first-strand synthesis system (Thermo Fisher Scientific). For quantification of mRNA levels, aliquoted cDNA was amplified with specific primers and PowerUp SYBR Master Mix (BioRad) by CFX384 Touch Real-Time PCR detection system (BioRad). Primers were optimized and designed to hybridize with different exons. Primers are listed here (name and sequence 5′ → 3′ are indicated).

> pla2g7 F: TCAAACTGCAGGCGCTTTTC, pla2g7 R: AGTACAAACGCACGAAGACG
> Egfr F: GCCATCTGGGCCAAAGATACC, Egfr: GTCTTCGCATGAATAGGCCAAT
> Lfng F: AAGATGGCTGTGGAGTATGACC, Lfng R: TCACTTTGTGCTCGCTGATC
> Ccn1a F: GATACCTGCTCGGGGAAAGAG, Ccn1a R: GCATTGGGGAAACTGTGTTGA
> Klf9 F: AAACACGCCTCCGAAAAGAG, Klf9 R: AACTGCTTTTCCCCAGTGTG
> Bmp4 F: GACCAGGTTCATTGCAGCTTTC, Bmp4 R: AAACGACCATCAGCATTCGG
> Actb F: CATTGCTGACAGGATGCAGAAGG, Actb R: TGCTGGAAGGTGGACAGTGAGG

## Statistical analysis

Statistical analysis was carried out using GraphPad Prism software. Both data collection and quantification were performed in a blinded manner. Data in figure panels reflect several independent experiments performed on different days. An estimate of variation within each group of data is indicated using standard error of the mean. Comparison of two groups was performed using two-tailed Student's unpaired *t*-test unless otherwise specified. Comparison of one group across time was performed using a one-way ANOVA with repeated measure. Comparison of two groups across treatment condition or time was performed using a two-way repeated measure ANOVA and main effects or interactions were followed by Bonferroni post hoc analysis. In the text and figure legends, '*n*' indicates number of mice per group. Detailed statistical analyses can be found in *Supplementary file 4*. For statistical analysis of DEGs, please see RNA-seq analysis section for details.

Two-photon imaging: In order to compare differences in the modes of RGL division between the two genotypes, we used the R statistical analysis software to fit a generalized linear mixed effects model to the division numbers across different mice, using genotype as a fixed effect, and including animal identity as a random effect in order to account for differences between individual animals [DivisionType ~ Genotype + (1|MouseIdentity)]. p values were calculated with a likelihood-ratio test comparing our model to a null model with no genotype information and identical random effects [DivisionType ~ 1 + (1|MouseIdentity)].

| Antibodies | Source | Identifier |
|---|---|---|
| Rat anti-BrdU | BioRad | Cat# MCA2483T, RRID:AB_1055584 |
| Rabbit anti-GFAP | Millipore | Cat# AB5804, RRID:AB_2109645 |
| Chicken anti-Nestin | Aves lab | Cat# NES, RRID:AB_2314882 |
| Rabbit anti-RFP | Rockland | Cat# 600-401-379, RRID:AB_2209751 |
| Chicken anti-GFAP | Millipore | Cat# AB5541, RRID:AB_177521 |
| Goat anti-GFP | Novus | NB100-1770, RRID:AB_10128178 |
| Goat anti-DCX | Santa Cruz Biotechnology | Cat# sc-8066, RRID:AB_2088494 |
| Mouse anti-beta galactosidase | Promega | Cat# Z3781, RRID:AB_430877 |
| Chicken anti-GFP | Abcam | Cat# ab13970, RRID:AB_300798 |
| Mouse anti-Mcm2 | BD Biosciences | Cat# 610700, RRID:AB_2141952 |
| NICD, rabbit anticleaved Notch1 | Assay Biotech | Cat# L0119, RRID:AB_10687460 |

*Continued on next page*

*Continued*

| Antibodies | Source | Identifier |
|---|---|---|
| Rabbit anti-HA | Cell Signaling | Cat# 3724, RRID:AB_1549585 |
| Anti-digoxigenin Fab fragments Antibody, POD conjugated | Roche | Cat# 11207733910, RRID:AB_514500 |
| Anti-digoxigenin Fab fragments Antibody, AP conjugated | Roche | Cat# 11093274910, RRID:AB_514497 |
| Alexa Fluor 488-, Cy3-, or Cy5-conjugated donkey secondary | Jackson ImmunoResearch | N/A |
| Goat anti-RFP | Sicgen | Cat# AB1140-100, RRID:AB_2877097 |

# Acknowledgements

We wish to thank members of Sahay and Goncalves labs for input on this work. NG received support from Department of Psychiatry, MGH. KM is a trainee in the Einstein Training Program in Stem Cell Research, supported by the Empire State Stem Cell Fund through New York State Department of Health Contract C34874GG. YS is recipient of a MGH ECOR Fund for Medical Discovery (FMD) Fundamental Research Fellowship Award. DG, CH, JC, and AZ are recipients of HSCI summer internship fellowships. AS acknowledges NINDS R56NS117529, Ellison Family Philanthropic support and the James and Audrey Foster MGH Research Scholar Award for supporting this work. JTG acknowledges support from US National Institutes of Health NINDS R56NS117529 and the Whitehall Foundation. AS thanks LMS Sahay for proof reading manuscript.

# Additional information

## Funding

| Funder | Grant reference number | Author |
|---|---|---|
| National Institute of Neurological Disorders and Stroke | R56NS117529 | J Tiago Gonçalves Amar Sahay |
| National Institutes of Health | NINDS R56NS117529 | J Tiago Gonçalves Amar Sahay |
| Whitehall Foundation | | J Tiago Gonçalves |
| James and Audrey Foster MGH Research Scholar Award | | Amar Sahay |
| Ellison Family Philanthropic support | | Amar Sahay |

The funders had no role in study design, data collection and interpretation, or the decision to submit the work for publication.

## Author contributions

Nannan Guo, Kelsey D McDermott, Data curation, Formal analysis, Investigation, Methodology, Validation, Visualization, Writing – review and editing; Yu-Tzu Shih, Formal analysis, Investigation, Methodology, Validation, Visualization, Writing – review and editing; Haley Zanga, Debolina Ghosh, Charlotte Herber, William R Meara, Alexia Zagouras, Formal analysis, Investigation, Methodology; James Coleman, Investigation, Methodology; Lai Ping Wong, Ruslan Sadreyev, Data curation, Investigation, Methodology; J Tiago Gonçalves, Data curation, Funding acquisition, Investigation, Methodology, Project administration, Resources, Supervision, Validation, Visualization, Writing – review

and editing; Amar Sahay, Conceptualization, Funding acquisition, Investigation, Methodology, Project administration, Resources, Supervision, Validation, Visualization, Writing - original draft, Writing – review and editing

### Author ORCIDs
Alexia Zagouras ⓘ http://orcid.org/0000-0003-0899-0910
Amar Sahay ⓘ http://orcid.org/0000-0003-0677-1693

### Ethics
Animals were handled and experiments were conducted in accordance with procedures approved by the Institutional Animal Care and Use Committee (IACUC) at the Massachusetts General Hospital (2011N000084 ) and Albert Einstein College of Medicine in accordance with NIH guidelines.

### Decision letter and Author response
Decision letter https://doi.org/10.7554/eLife.72195.sa1
Author response https://doi.org/10.7554/eLife.72195.sa2

---

## Additional files

### Supplementary files
• Supplementary file 1. Complete lists of differentially expressed genes (DEGs) in Gli1+ radial-glial neural stem cells (RGLs) following Kruppel-like factor 9 (*Klf9*) deletion. DEGs were defined by at least 1.2-fold change with FDR < 0.05.

• Supplementary file 2. Gene ontology annotation (gGOSt, https://biit.cs.ut.ee/gprofiler/gost) of differentially upregulated genes in Gli1+ radial-glial neural stem cells (RGLs) following Kruppel-like factor 9 (*Klf9*) deletion.

• Supplementary file 3. Gene ontology annotation (gGOSt, https://biit.cs.ut.ee/gprofiler/gost) of differentially downregulated genes in Gli1+ radial-glial neural stem cells (RGLs) following Kruppel-like factor 9 (*Klf9*) deletion.

• Supplementary file 4. Statistical Analysis.

• Transparent reporting form

### Data availability
Sequencing data have been deposited in GEO under accession code GSE164889.

The following dataset was generated:

| Author(s) | Year | Dataset title | Dataset URL | Database and Identifier |
|---|---|---|---|---|
| Guo N, Sahay A | 2021 | Transcriptional regulation of neural stem cell expansion in adult hippocampus | http://www.ncbi.nlm.nih.gov/geo/query/acc.cgi?acc=GSE164889 | NCBI Gene Expression Omnibus, GSE164889 |

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
