## [Editor Report]

In this study, Guo et al. uncover a role for the transcription factor Klf9 in keeping adult hippocampal neural stem cells in a state of quiescence. Following relief from this molecular brake through Klf9 loss-of-function, neural stem cells undergo symmetric cell divisions that promote their self-renewal and expansion. This data suggest that Klf9 contributes to the molecular interplay that governs stem cell decisions between quiescence and activation on one hand and between distinct modes of cell divisions on the other.

---

## [Decision Letter]

**Decision letter after peer review:**

Thank you for submitting your article "Transcriptional regulation of neural stem cell expansion in adult hippocampus" for consideration by *eLife*. Your article has been reviewed by 2 peer reviewers, and the evaluation has been overseen by Joseph Gleeson as the Reviewing Editor and Marianne Bronner as the Senior Editor. The following individual involved in review of your submission has agreed to reveal their identity: Benedikt Berninger (Reviewer #1).

Overall the reviewers were positive on the findings, and have suggested a revision based upon their comments. The reviewers have also discussed their reviews with one another, and the Reviewing Editor has drafted this to help you prepare a revised submission.

Essential revisions:

1) The authors showed that Klf9 promoter is more active in quiescent RGLs (using LacZ reporter) and that Klf9 transcripts are more abundant in these cells compared to activated RGLs. But they did not show the expression pattern of Klf9 proteins. Further co-immunostaining studies of Klf9 and neurogenic cell stage-specific markers are needed to determine in which cell stage Klf9 protein is expressed and whether Klf9 protein levels are higher in quiescent RGLs compared to those in activated RGLs.

2) Knockout efficiency of Klf9 (transcript and/or protein) should be determined in each of the various tamoxifen injection paradigms to support the validity of the knockout model.

3) demonstrating how much Klf9 protein is overexpressed in RGLs of the Sox1 tTA; teto Klf9 mice will strengthen the conclusion that Klf9 overexpression suppresses RGL activation.

*Reviewer 1:*

The study by Sahay and colleagues addressed the role of Klf9 in radial glia like neural stem cells (RGLs) in the adult dentate gyrus. By gain- and loss-of-function experiments, they provide evidence that higher levels of Klf9 normally maintain RGLs in a quiescent state. By inducing conditional loss-of-function in a RGL-specific Gli1 Cre^ERT2^ driver line, the authors found that RGL enter cell cycle at higher rate as controls. Clonal analyses as well as longitudinal in vivo live imaging provide support for the notion that following loss of Klf9 RGLs undergo symmetric self-renewing cell divisions. By analysing the translatome of control and Klf9-deficient RGLs, the authors find that absence of Klf9 promotes the expression of transcripts involved in stem cell self-renewal, while transcripts related to maintaining stem cell quiescence become downregulated. Overall, the authors conclude that Klf9 acts as a brake on symmetric self-renewal of RGLs.

Both, in vivo clonal analysis and longitudinal live imaging used to demonstrate the increase in self-renewing RGL divisions are technically challenging. Thus, for the untrained eye, some of the image sequences are difficult to interpret. In contrast, the Ribotag experiments provide a clear picture consistent with the interpretation of the authors that loss of Klf9 promotes cell cycle entry and loss of quiescence. This is very exciting as very little is known to which degree such symmetric cell divisions still occur in adult RGLs, and even less how these are regulated. Through the identification of Klf9 as a counter-player to other transcription factors promoting neurogenic lineage progression, the study by Guo et al. uncovers a new layer of molecular regulation of the intricate balance between stem cell quiescence, self-renewal and differentiation.

1) Ideally, the authors should provide evidence for successful recombination of the conditional Klf9 locus when crossing with the Gli1 Cre^ERT2^ driver line. POMC-Cre is not a perfect surrogate as Cre expression and Cre activity may be very different between these driver lines and hence result in distinct recombination efficiencies. The experiment using POMC-Cre lines essentially proves that the Klf9 can be recombined, but not that it does become recombined in Gli1 Cre^ERT2^ drivers.

2) S2 could be a main figure. However, one would like to see examples images for Mcm2 and BrdU for both conditions (control vs Klf9 gain of function). What's the evidence for Klf9 over expression?

How was the cell identity assignment in Figure 2C performed?

3) Live imaging experiments are certainly very challenging. However, some of the video stills are difficult to interpret. In 3B, second row, I fail to see the second cell. I suppose it is due to the difficulty to discern individual cells that the authors refrained from generating lineage trees. By the way, I wasn't able to identify whether the examples shown in 3B represent KLf9 deficient or control RGLs. To clarify what can be seen in the videos, would it be possible to generate narrated video files that make any observation more explicit?

*Reviewer #2:*

Guo, McDermott et al. examined the role of the transcription factor Klf9 in mouse adult hippocampal neural stem cells, i.e., radial glia-like cells (RGLs). They first show that Klf9 gene expression is higher in quiescent RGLs compared to dividing RGLs. Using clonal analysis and longitudinal intravital imaging, they find that deletion of Klf9 leads to increased number of RGL clones. Transcriptomic studies with Ribotag provide insight into possible mechanisms by which Klf9 regulates symmetric self-renewal of RGLs. Overall, this paper identifies Klf9 as a critical regulator of RGL self-renewal and increases our understanding of transcriptional regulation of adult hippocampal neurogenesis.

Strengths:

A significant strength of the paper is that they used two different approaches to examine the consequences of Klf9 deletion on RGL cell proliferation. The first approach is clonal analysis by sparsely labeling adult hippocampal RGLs with a fluorescent reporter. Comparison of labeled clones between control and Klf9 knockout mice reveals that mutant mice contain more clones with 2 or more RGLs, suggesting that deletion of Klf9 increases symmetric cell division of RGLs. In the second approach, they induced fluorescent reporter expression in RGLs of control and Klf9 knockout mice and followed their cell divisions using intravital 2-photon imaging. This analysis shows that, while knockout RGLs undergo similar rounds of cell division as control RGLs, more knockout RGLs undergo symmetric cell division. Altogether, these two independent and complementary approaches provide strong evidence that Klf9 normally suppresses symmetric self-renewal of RGLs.

Weaknesses:

Although the main claim that Klf9 regulates RGL self-renewal is largely supported by the clonal analysis and longitudinal intravital imaging, additional analyses are needed to fully support this claim. An alternative approach is also necessary to better understand the mechanism by which Klf9 regulates RGL self-renewal.

1) The authors showed that Klf9 promoter is more active in quiescent RGLs (using LacZ reporter) and that Klf9 transcripts are more abundant in these cells compared to activated RGLs. But they did not show the expression pattern of Klf9 proteins. Further co-immunostaining studies of Klf9 and neurogenic cell stage-specific markers are needed to determine in which cell stage Klf9 protein is expressed and whether Klf9 protein levels are higher in quiescent RGLs compared to those in activated RGLs.

2) This paper uses the Gli1-Cre^ERT2^ to conditionally delete Klf9 from adult hippocampal stem and progenitor cells. But the knockout efficiencies of Klf9 in the various tamoxifen injection paradigms remain unexamined. Rather, the authors rely solely on the tdTomato reporter expression. While tdTomato expression is indicative of Cre activity, it does not necessarily correlate with the knockout efficiency of Klf9. Further illustrating this concern, the Ribotag transcriptomic study shows only ~60% reduction in Klf9 transcript levels in the knockout mice, which suggests incomplete Klf9 deletion in tdTomato+ cells and/or contamination of non-knockout cells during RNA isolation. Therefore, the knockout efficiency of Klf9 (transcript and/or protein) should be determined in each of the various tamoxifen injection paradigms to support the validity of the knockout model. Along these lines, demonstrating how much Klf9 protein is overexpressed in RGLs of the Sox1 tTA; teto Klf9 mice will strengthen the conclusion that Klf9 overexpression suppresses RGL activation.

3) To investigate gene expression changes upon Klf9 knockout and to understand the mechanism by which Klf9 regulates RGL self-renewal, the authors performed transcriptomic studies using the Ribotag technique. While the Ribotag approach can reduce cell stress response due to single cell isolation (as the authors discuss), a major concern with this technique in this case is that the isolated RNAs come from a mixture of RGLs (quiescent and activated) and their neuronal progenies, as well as astrocytes. Most of the notable DEGs that they mention are expressed in RGLs, astrocytes, and other neuronal lineage cell stages. It is unclear whether the changes of genes and pathways occur in RGLs or their progenies or astrocytes. It is also impossible to tease apart the expression changes between quiescent and activated RGLs with this approach. Finally, whether Klf9 directly regulates the expression of these genes remains unknown. For these reasons, it is difficult to interpret the transcriptomic data and how gene expression changes may contribute to altered RGL self-renewal. Single-cell RNAseq (and maybe single-cell ATAC-seq) studies of tdTomato+ cells from control and Klf9 knockout mice will allow the authors to address these concerns.

Here are additional experiments and edits that will help improve the clarity of the paper and strengthen the conclusions.

1) Many of the photomicrographs are not of sufficient quality to clearly support the conclusions. For example, Figure 1B could include insets of higher magnification of the cells of interest; Figure 1F could include representative images of lower magnification (e.g. a cross section of the entire dentate gyrus) of control and knockout mice to show that the observed increase in activated RGLs is consistent across the subgranular zone; it is extremely difficult to distinguish RGLs, neuronal progenies and astrocytes in Figure 2C; the increase in NICD in knockout RGLs is questionable as the IF pictures presented in Figure 4E do not allow for clear identification of RGLs.

2) The "R2" cell in Figure S3B appears to lack a single strong radial process, raising a concern about the identity of this cell and, more generally, the correct identification of RGLs, RGL progenies, and astrocytes in the in vivo imaging studies. One suggestion is to include nestin as an additional marker for RGL identification. It is also unclear whether post-hoc IF studies were done to verify all cells of the in vivo imaging study, or only a subset of cells was verified histologically.

3) The authors identified activated RGLs in control and Sox1 tTA; teto Klf9 mice using MCM2+NES+ (Figure S2B). Since NES is also expressed in pericytes and endothelial cells, both of which are capable of cell division, it will be important to include an additional RGL marker such as GFAP or *SOX2*.

4) In the experiment presented in Figure S2C, the authors injected BrdU for 14 days and examined the fraction of BrdU^+^NES+ RGLs, which they interpreted as "activated and dividing cells". However, at the time of sampling, which is 14 days post the initial BrdU injection, BrdU^+^ cells are a mixture of cells that are actively dividing (recent BrdU uptake) or cells that are no longer dividing (e.g. BrdU uptake occurred days ago). Therefore, BrdU^+^ cells should not be interpreted as "activated and dividing cells" in this 14-day labeling study. A short-term (less than a few hours) BrdU labeling study will be more appropriate to address the effect of Klf9 overexpression on RGL cell activation.

5) There are a couple of formatting issues: mouse alleles are sometimes written as, for example, "Gli1Cre^ERT2^:Klf9f/f", and sometimes as ""Gli1Cre^ERT2^;Klf9f/f". Please be consistent. A couple of gene names are not accurate: Igfbp has multiple members, so it is unclear which Igfbp it is. Gene symbol for cyclin A1 is Cdkn1a not Ccn1a (Figure 4D).

6) There is one piece of data mentioned in the results but not shown. Please show all data described in the results.

7) In the method of "clonal lineage analysis", the author wrote "all the labelled cells within one clone were in close spatial proximity to each other". It will be important to provide a distance cutoff for this analysis.

8) The dose of TAM is missing in Figure 4A.

---

## [Author Response]

Essential revisions:1) The authors showed that Klf9 promoter is more active in quiescent RGLs (using LacZ reporter) and that Klf9 transcripts are more abundant in these cells compared to activated RGLs. But they did not show the expression pattern of Klf9 proteins. Further co-immunostaining studies of Klf9 and neurogenic cell stage-specific markers are needed to determine in which cell stage Klf9 protein is expressed and whether Klf9 protein levels are higher in quiescent RGLs compared to those in activated RGLs.

Our *Klf9* fluorescent in situ hybridization data shows higher levels of *Klf9* expression in quiescent neural stem cells relative to activated neural stem cells. Importantly, this riboprobe is validated using *Klf9* null and *Klf9* cKO tissue. This experimental data is also independently corroborated by single cell RNA sequencing data published in Bottes et al. [1](Jessberger, personal communication). We are also very interested in visualizing Klf9 protein levels in different cell-types and during different stages of maturation. Unfortunately, all available antibodies against Klf9 do not work for immunohistochemistry as ascertained using *Klf9* KO tissue as controls. Given these challenges we have generated a mCherry-Klf9 (N terminus fusion) knock-in mouse line, but this reagent is still under characterization. If after thorough characterization and validation, this mouse line is deemed useful, we will make this reagent available to the neuroscience and stem cell research communities through JAX.

2) Knockout efficiency of Klf9 (transcript and/or protein) should be determined in each of the various tamoxifen injection paradigms to support the validity of the knockout model.

Thank you for raising this question. In Figure 1-supplement 2, we provide experimental evidence for estimating recombination efficiency of *Klf9* in Gli1-positive tdTomato labeled neural stem cells. Using a TAM induction protocol that was used for in vivo longitudinal twophoton imaging, we found a 32% reduction in *Klf9* transcript associated fluorescence intensity in Gli-1 positive neural stem cells. This data validates our cKO model and also suggests that our analysis of impact of *Klf9* recombination in Gli1-positive NSCs on division mode is likely to reflect an underestimate at best.

Unfortunately, we cannot compare the effect of *Klf9* deletion in neural stem cells with that of other genes that regulate neural stem cell activation and division mode since most published studies do not estimate recombination frequencies of target alleles [For eg: [2] [3] [4] [5] but see [6] which shows IHC but does not have quantification].

3) Demonstrating how much Klf9 protein is overexpressed in RGLs of the Sox1 tTA; teto Klf9 mice will strengthen the conclusion that Klf9 overexpression suppresses RGL activation.

We (or for that matter anybody in the published domain) do not have antibodies against Klf9 that are useful for immunohistochemistry. Unfortunately, we are unable to breed the old Sox1tTA male animals with tet0Klf9/tet0 Klf9 female mice to generate new tissue for any further analysis. When we obtained the original breeders almost 6-7 years ago from Dr. Robert Blelloch (University of California, San Fransisco), it took us almost a year to generate animals carrying tTA and tet0 Klf9 alleles.

Since this Sox1tTA mouse line targets both neural stem cells and progenitors it offers circumstantial evidence in support of our main claim regarding Klf9 functions in regulation of neural stem cell activation. Indeed, we are very careful in our wording in the manuscript to reflect the lack of neural stem cell specificity of the Sox1tTA line. In contrast, the genetic evidence obtained from Gli1Cre^ERT2^:*Klf9*^f/f or +/+^;Ai14 mice is the most compelling since the Gli1Cre^ERT2^driver line targets neural stem cells and not progenitors. We now provide a second line of direct evidence implicating Klf9 as a brake on activation of neural stem cells using Ascl1Cre^ERT2^:*Klf9*^f/f or +/+^;Ai14 mice and this data is shown in Author response image 1. The Ascl1 Cre^ERT2^ mouse line targets a distinct population of neural stem cells that is biased towards asymmetric neurogenic divisions, exhibits short-term self-renewal and division-coupled depletion [1]. We found that conditional deletion of *Klf9* in Ascl1 Cre^ERT2^ targeted adult hippocampal RGLs significantly increased the fraction of activated RGLs (% of MCM2+tdTomato+RGLs).

**Author response image 1. sa2fig1:** Inducible deletion of *Klf9* in Ascl1+ RGLs in adult mice (Ascl1 Cre^ERT2^*:Klf9*^+/+^:Ai14 vs. Ascl1 Cre^ERT2^*:Klf9*^f/f^:Ai14) results in increased RGL activation (percentage of MCM2+tdTomato+Nestin+RGLs) n=4, 5 mice/group. Data are represented as mean ± SEM. * p=0.02, Scale Bar 50 µm.

Given the specificity of our genetic deletion experiments using Gli1 Cre^ERT2^ and Ascl1 Cre^ERT2^ lines we argue that we have sufficient evidence to support the claim that “*Klf9* deletion in adult hippocampal RGLs increases activation”. We are happy to remove the Sox1 tTA dataset if you would like us to do so.

Reviewer 1:The study by Sahay and colleagues addressed the role of Klf9 in radial glia like neural stem cells (RGLs) in the adult dentate gyrus. By gain-and loss-of-function experiments, they provide evidence that higher levels of Klf9 normally maintain RGLs in a quiescent state. By inducing conditional loss-of-function in a RGL-specific Gli1 Cre^ERT2^ driver line, the authors found that RGL enter cell cycle at higher rate as controls. Clonal analyses as well as longitudinal in vivo live imaging provide support for the notion that following loss of Klf9 RGLs undergo symmetric self-renewing cell divisions. By analysing the translatome of control and Klf9-deficient RGLs, the authors find that absence of Klf9 promotes the expression of transcripts involved in stem cell self-renewal, while transcripts related to maintaining stem cell quiescence become downregulated. Overall, the authors conclude that Klf9 acts as a brake on symmetric self-renewal of RGLs.Both, in vivo clonal analysis and longitudinal live imaging used to demonstrate the increase in self-renewing RGL divisions are technically challenging. Thus, for the untrained eye, some of the image sequences are difficult to interpret. In contrast, the Ribotag experiments provide a clear picture consistent with the interpretation of the authors that loss of Klf9 promotes cell cycle entry and loss of quiescence. This is very exciting as very little is known to which degree such symmetric cell divisions still occur in adult RGLs, and even less how these are regulated. Through the identification of Klf9 as a counter-player to other transcription factors promoting neurogenic lineage progression, the study by Guo et al. uncovers a new layer of molecular regulation of the intricate balance between stem cell quiescence, self-renewal and differentiation.1) Ideally, the authors should provide evidence for successful recombination of the conditional Klf9 locus when crossing with the Gli1 Cre^ERT2^ driver line. POMC-Cre is not a perfect surrogate as Cre expression and Cre activity may be very different between these driver lines and hence result in distinct recombination efficiencies. The experiment using POMC-Cre lines essentially proves that the Klf9 can be recombined, but not that it does become recombined in Gli1 Cre^ERT2^ drivers.

In Figure 1-supplement 2, we provide experimental evidence for estimating recombination efficiency of *Klf9* in Gli1-positive tdTomato labeled neural stem cells. Using a TAM induction protocol that was used for in vivo longitudinal two-photon imaging, we found a 32% reduction in *Klf9* transcript associated fluorescence intensity in Gli-1 positive neural stem cells. This data validates our cKO model and also suggests that our analysis of impact of *Klf9* recombination in Gli1-positive NSCs on division mode is likely to reflect an underestimate at best. Unfortunately, we cannot compare the effect of *Klf9* deletion in neural stem cells with that of other genes that regulate neural stem cell activation and division mode since most published studies do not estimate recombination frequencies of target alleles [For eg: [2] [3] [4] [5] but see [6] which shows IHC but does not have quantification].

2) S2 could be a main figure. However, one would like to see examples images for Mcm2 and BrdU for both conditions (control vs Klf9 gain of function). What's the evidence for Klf9 over expression?

Sorry for this omission. As requested, we have added images for MCM2 and Brdu labeling for Sox1 tTA:tet0 Klf9/Klf9 mice.

As stated earlier we (or for that matter anybody in the published domain) do not have antibodies against Klf9 that are useful for immunohistochemistry. Unfortunately, we are unable to breed the old Sox1tTA male animals with tet0Klf9/tet0 Klf9 female mice to generate new tissue for any further analysis. When we obtained the original breeders almost 6-7 years ago from Dr. Robert Blelloch (University of California, San Fransisco), it took us almost a year to generate animals carrying tTA and tet0 Klf9 alleles.

Since this Sox1tTA mouse line targets both neural stem cells and progenitors it offers circumstantial evidence in support of our main claim regarding Klf9 functions in regulation of neural stem cell activation. Indeed, we are very careful in our wording in the manuscript to reflect the lack of neural stem cell specificity of the Sox1tTA line. In contrast, the genetic evidence obtained from Gli1Cre^ERT2^:*Klf9*^f/f or +/+^;Ai14 mice is the most compelling since the Gli1Cre^ERT2^driver line targets neural stem cells and not progenitors. We now provide a second line of direct evidence implicating Klf9 as a brake on activation of neural stem cells using Ascl1Cre^ERT2^:*Klf9*^f/f or +/+^;Ai14 mice and this data is shown in Author response image 1. The Ascl1 Cre^ERT2^ mouse line targets a distinct population of neural stem cells that is biased towards asymmetric neurogenic divisions, exhibits short-term self-renewal and division-coupled depletion [1]. We found that conditional deletion of *Klf9* in Ascl1 Cre^ERT2^ targeted adult hippocampal RGLs significantly increased the fraction of activated RGLs (% of MCM2+tdTomato+RGLs).

Given the specificity of our genetic deletion experiments using Gli1 Cre^ERT2^ and Ascl1 Cre^ERT2^ lines we argue that we have sufficient evidence to support the claim that “*Klf9* deletion in adult hippocampal RGLs increases activation”. We are happy to remove the Sox1 tTA dataset if you would like us to do so.

How was the cell identity assignment in Figure 2C performed?

Confocal analysis of td Tomato labeled RGLs (distinct radial glial like morphology) that are labeled for GFAP. We now provide additional analysis in Figure 2—figure supplement 1 and Videos 1-8 to clearly convey clonal compositions.

3) Live imaging experiments are certainly very challenging. However, some of the video stills are difficult to interpret. In 3B, second row, I fail to see the second cell. I suppose it is due to the difficulty to discern individual cells that the authors refrained from generating lineage trees. By the way, I wasn't able to identify whether the examples shown in 3B represent KLf9 deficient or control RGLs. To clarify what can be seen in the videos, would it be possible to generate narrated video files that make any observation more explicit?

Yes, we have generated Narrated videos (Video 9-10). Thank you for this suggestion.

Reviewer #2:[…]1) The authors showed that Klf9 promoter is more active in quiescent RGLs (using LacZ reporter) and that Klf9 transcripts are more abundant in these cells compared to activated RGLs. But they did not show the expression pattern of Klf9 proteins. Further co-immunostaining studies of Klf9 and neurogenic cell stage-specific markers are needed to determine in which cell stage Klf9 protein is expressed and whether Klf9 protein levels are higher in quiescent RGLs compared to those in activated RGLs.

Our *Klf9* fluorescent in situ hybridization data shows higher levels of *Klf9* expression in quiescent neural stem cells relative to activated neural stem cells. Importantly, this riboprobe is validated using *Klf9* null and cKO tissue. This experimental data is also independently corroborated by single cell RNA sequencing data published in Bottes et al. [1](Jessberger, personal communication). We are also very interested in visualizing Klf9 protein levels in different cell-types and during different stages of maturation. Unfortunately, all available antibodies against Klf9 do not work for immunohistochemistry as ascertained using *Klf9* KO tissue as controls. Given these challenges we have generated a mCherry-Klf9 (N terminus fusion) knock-in mouse line, but this reagent is still under characterization. If after thorough characterization and validation this mouse line is deemed useful, we will make this reagent available to the neuroscience and stem cell research communities through JAX.

2) This paper uses the Gli1-Cre^ERT2^ to conditionally delete Klf9 from adult hippocampal stem and progenitor cells. But the knockout efficiencies of Klf9 in the various tamoxifen injection paradigms remain unexamined. Rather, the authors rely solely on the tdTomato reporter expression. While tdTomato expression is indicative of Cre activity, it does not necessarily correlate with the knockout efficiency of Klf9. Further illustrating this concern, the Ribotag transcriptomic study shows only ~60% reduction in Klf9 transcript levels in the knockout mice, which suggests incomplete Klf9 deletion in tdTomato+ cells and/or contamination of non-knockout cells during RNA isolation. Therefore, the knockout efficiency of Klf9 (transcript and/or protein) should be determined in each of the various tamoxifen injection paradigms to support the validity of the knockout model. Along these lines, demonstrating how much Klf9 protein is overexpressed in RGLs of the Sox1 tTA; teto Klf9 mice will strengthen the conclusion that Klf9 overexpression suppresses RGL activation.

In Figure 1-supplement 2, we provide new experimental evidence for estimating recombination efficiency of *Klf9* in Gli1-positive tdTomato labeled neural stem cells. Using a TAM induction protocol that was used for in vivo longitudinal two-photon imaging, we found a 32% reduction in *Klf9* transcript associated fluorescence intensity in Gli-1 positive neural stem cells. This data validates our cKO model and also suggests that our analysis of impact of *Klf9* recombination in Gli1-positive NSCs on division mode is likely to reflect an underestimate at best. Unfortunately, we cannot compare the effect of *Klf9* deletion in neural stem cells with that of other genes that regulate neural stem cell activation and division mode since most published studies do not estimate recombination frequencies of target alleles [For eg: [2] [3] [4] [5] but see [6] which shows IHC but does not have quantification].

As stated earlier we (or for that matter anybody in the published domain) do not have antibodies against Klf9 that are useful for immunohistochemistry. Unfortunately, we are unable to breed the old Sox1tTA male animals with tet0Klf9/tet0 Klf9 female mice to generate new tissue for any further analysis. When we obtained the original breeders almost 6-7 years ago from Dr. Robert Blelloch (University of California, San Fransisco), it took us almost a year to generate animals carrying tTA and tet0 Klf9 alleles.

Since this mouse line targets both neural stem cells and progenitors it offers circumstantial evidence in support of our main claim regarding Klf9 functions in regulation of neural stem cell activation. Indeed, we are very careful in our wording in the manuscript to reflect the lack of neural stem cell specificity of the Sox1tTA line. In contrast, the genetic evidence obtained from Gli1Cre^ERT2^:*Klf9*^f/f or +/+^;Ai14 mice is the most compelling since the Gli1Cre^ERT2^driver line targets neural stem cells and not progenitors. We now provide a second line of direct evidence implicating Klf9 as a brake on activation of neural stem cells using Ascl1Cre^ERT2^:*Klf9*^f/f or +/+^;Ai14 mice and this data is shown in Author response image 1. The Ascl1 Cre^ERT2^ mouse line targets a distinct population of neural stem cells that is biased towards asymmetric neurogenic divisions, exhibits short-term self-renewal and division-coupled depletion [1]. We found that conditional deletion of *Klf9* in Ascl1 Cre^ERT2^ targeted adult hippocampal RGLs significantly increased the fraction of activated RGLs (% of MCM2+tdTomato+RGLs).

Given the specificity of our genetic deletion experiments using Gli1 Cre^ERT2^ and Ascl1 Cre^ERT2^ lines we argue that we have sufficient evidence to support the claim that “*Klf9* deletion in adult hippocampal RGLs increases activation”. We are happy to remove the Sox1 tTA dataset if you would like us to do so.

3) To investigate gene expression changes upon Klf9 knockout and to understand the mechanism by which Klf9 regulates RGL self-renewal, the authors performed transcriptomic studies using the Ribotag technique. While the Ribotag approach can reduce cell stress response due to single cell isolation (as the authors discuss), a major concern with this technique in this case is that the isolated RNAs come from a mixture of RGLs (quiescent and activated) and their neuronal progenies, as well as astrocytes. Most of the notable DEGs that they mention are expressed in RGLs, astrocytes, and other neuronal lineage cell stages. It is unclear whether the changes of genes and pathways occur in RGLs or their progenies or astrocytes. It is also impossible to tease apart the expression changes between quiescent and activated RGLs with this approach. Finally, whether Klf9 directly regulates the expression of these genes remains unknown. For these reasons, it is difficult to interpret the transcriptomic data and how gene expression changes may contribute to altered RGL self-renewal. Single-cell RNAseq (and maybe single-cell ATAC-seq) studies of tdTomato+ cells from control and Klf9 knockout mice will allow the authors to address these concerns.

(i) Every approach has its unique strengths. To the best of our knowledge, this is the first study to date that has performed ribosomal profiling of adult hippocampal neural stem cells to analyze the impact of gene deletion on gene expression in vivo. The experimental protocol took over 6 months to optimize given the low density of Gli-1 positive neural stem cells in the adult dentate gyrus. We chose the Gli1 Cre^ERT2^ driver line because it is restricts Cre^ERT2^ mediated recombination to a largely quiescent subpopulation of neural stem cells and not in progenitors. This is not true for Ascl1 or Cre^ERT2^ driver lines which target both neural stem cells and progenitors. (ii) scRNA sequencing of (Gli1 positive) adult hippocampal neural stem cells following inducible gene deletion in this sparse cell population has not been done to date. (iii) The ribosomal profiling approach minimizes stress which is a shortcoming of standard scRNA sequencing methodology. (iv) There are several lines of evidence that support the thesis that our DEGs reflect changes in neural stem cell properties. First, we observe an enrichment of genes expressed exclusively in RGLs (vs progenitors or neurons) consistent with changes in RGL numbers driven by division mode. And when you look at our database, there are many such “RGL only” genes (Pla2g7, Mlc1 etc). Second, the clear signature of increased FAO is consistent with a large and growing set of data suggesting that FAO is required to maintain non-differentiating /proliferative state. Third, we validated potentiation of Notch signaling by IHC for NICD in RGLs.

We sincerely hope that our dataset will serve as a resource for the stem cell research community to pursue candidate genes that regulate symmetric self-renewal of somatic stem cells. Of course, future studies relying on scRNA sequencing will further edify the extent to which our framework is valid.

Re-Klf9 and direct regulation of target genes and scATAC seq: Yes, we are very interested in these experiments. But these experiments are well beyond the scope of the current study.

Here are additional experiments and edits that will help improve the clarity of the paper and strengthen the conclusions.1) Many of the photomicrographs are not of sufficient quality to clearly support the conclusions. For example, Figure 1B could include insets of higher magnification of the cells of interest; Figure 1F could include representative images of lower magnification (e.g. a cross section of the entire dentate gyrus) of control and knockout mice to show that the observed increase in activated RGLs is consistent across the subgranular zone; it is extremely difficult to distinguish RGLs, neuronal progenies and astrocytes in Figure 2C; the increase in NICD in knockout RGLs is questionable as the IF pictures presented in Figure 4E do not allow for clear identification of RGLs.

We now provide additional analysis of Figure 2C in Figure 2—figure supplement 1 and Videos 18 to clearly convey clonal compositions.

We furnish new images in Figure 4E that clearly capture the increase in NICD levels in tdTomato labeled Gli1-Positive RGLs following deletion of *Klf9*.

Author response image 2 is a lower magnification of DG of Gli1Cre^ERT2^:*Klf9*^f/f or +/+^;Ai14 mice where the reader can appreciate the increase in RGL numbers throughout the DG in this population analysis done at 7dpi. We induced *Klf9* recombination and tdTomato expression in RGLs of adult Gli1 Cre^ERT2^; *Klf9*^f/f or +/+^; Ai14 mice (TAM 250 mg/Kg) and processed brain sections for tdTomato and GFAP immunohistochemistry at 7 dpi. We found that cell-autonomous deletion of *Klf9* in Gli1+RGLs resulted in a significant increase in the total number of tdTomato labeled RGLs and the fraction of tdTomato labeled RGLs at 7dpi.

**Author response image 2. sa2fig2:** A. Inducible deletion of *Klf9* in Gli1+RGLs in adult mice (Gli1Cre^ERT2^*:Klf9*^+/+^:Ai14 vs. Gli1 Cre^ERT2^*:Klf9*^f/f^:Ai14) results in expansion of RGLs as assessed at 7dpi. Representative images (A-B) in and corresponding quantification (bottom) 7 dpi: n=4 and 5 mice/group. Data are represented as mean ± SEM. ** p<0.01. White arrowheads indicate RGLs. Note the dramatic expansion in RGL numbers throughout DG in B. Scale Bar is 20 µm.

This is also the dataset that we refer to in the Results section as “data not shown”. “Population level lineage tracing experiments at short-term chase time points suggested that Klf9 loss in Gli1+ RGLs increased RGL numbers (data not shown).” In fact, it was this population analysis dataset that gave us a first glimpse into the possibility that Klf9 acts as a brake on RGL expansion. Given the caveats of population analysis, we invested in state-of-the art gold standard approaches, clonal analysis and in vivo longitudinal two-photon imaging of single RGLs, to test this possibility.

2) The “R2” cell in Figure S3B appears to lack a single strong radial process, raising a concern about the identity of this cell and, more generally, the correct identification of RGLs, RGL progenies, and astrocytes in the in vivo imaging studies. One suggestion is to include nestin as an additional marker for RGL identification. It is also unclear whether post-hoc IF studies were done to verify all cells of the in vivo imaging study, or only a subset of cells was verified histologically.

Please see new images following Imaris processing in Figure 3—figure supplement 1 (bottom panel). Only a subset of cells was verified histologically and this is now stated in the text (Line 191).

3) The authors identified activated RGLs in control and Sox1 tTA; teto Klf9 mice using MCM2+NES+ (Figure S2B). Since NES is also expressed in pericytes and endothelial cells, both of which are capable of cell division, it will be important to include an additional RGL marker such as GFAP or SOX2.

Cells were identified as RGLs based on Nestin immunohistochemistry and characteristic radial glial-like morphology (apical process traversing granule cell layer) in subgranular zone and GFAP overlap. Please see new the Panel of images in Figure 1—figure supplement 3.

4) In the experiment presented in Figure S2C, the authors injected BrdU for 14 days and examined the fraction of BrdU^+^NES+ RGLs, which they interpreted as “activated and dividing cells”. However, at the time of sampling, which is 14 days post the initial BrdU injection, BrdU^+^ cells are a mixture of cells that are actively dividing (recent BrdU uptake) or cells that are no longer dividing (e.g. BrdU uptake occurred days ago). Therefore, BrdU^+^ cells should not be interpreted as “activated and dividing cells” in this 14-day labeling study. A short-term (less than a few hours) BrdU labeling study will be more appropriate to address the effect of Klf9 overexpression on RGL cell activation.

The reason why we performed 2 weeks of daily BrdU was to tag/label enough neural stem cells (which unlike neural progenitors are rarely dividing cells). “Less than a few hours” as you suggest will tag primarily and almost exclusively rapidly dividing progenitors and extremely few neural stem cells. You are correct that in our BrdU^+^Nestin RGLs analysis we also have some label retaining cells. However, please note that these cells must have divided in the first place to acquire BrdU. *Klf9* overexpression (prior to BrdU pulse) decreases this possibility thereby arguing for an anti-proliferative role for Klf9.

5) There are a couple of formatting issues: mouse alleles are sometimes written as, for example, “Gli1Cre^ERT2^:Klf9f/f”, and sometimes as “”Gli1Cre^ERT2^;Klf9f/f”. Please be consistent. A couple of gene names are not accurate: Igfbp has multiple members, so it is unclear which Igfbp it is. Gene symbol for cyclin A1 is Cdkn1a not Ccn1a (Figure 4D).

Changes made. CyclinA1 is Ccn1a.

6) There is one piece of data mentioned in the results but not shown. Please show all data described in the results.

We show this data (expansion of the RGL pool) in Author response image 2.

7) In the method of “clonal lineage analysis”, the author wrote “all the labelled cells within one clone were in close spatial proximity to each other”. It will be important to provide a distance cutoff for this analysis.

We now state this “A ring with a radius of 50 μm from the center of the RGL was used to determine the clone composition” in Methods.

8) The dose of TAM is missing in Figure 4A.

Done. Thank you for your comments

References:

1. Bottes, S., Jaeger, B.N., Pilz, G.A., Jorg, D.J., Cole, J.D., Kruse, M., Harris, L., Korobeynyk, V.I., Mallona, I., Helmchen, F., et al. (2020). Long-term self-renewing stem cells in the adult mouse hippocampus identified by intravital imaging. Nat Neurosci.

2. Bonaguidi, M.A., Wheeler, M.A., Shapiro, J.S., Stadel, R.P., Sun, G.J., Ming, G.L., and Song, H. (2011). in vivo clonal analysis reveals self-renewing and multipotent adult neural stem cell characteristics. Cell *145*, 1142-1155.

3. Song, J., Zhong, C., Bonaguidi, M.A., Sun, G.J., Hsu, D., Gu, Y., Meletis, K., Huang, Z.J., Ge, S., Enikolopov, G., et al. (2012). Neuronal circuitry mechanism regulating adult quiescent neural stem-cell fate decision. Nature *489*, 150-154.

4. Knobloch, M., von Schoultz, C., Zurkirchen, L., Braun, S.M., Vidmar, M., and Jessberger, S. (2014). SPOT14-positive neural stem/progenitor cells in the hippocampus respond dynamically to neurogenic regulators. Stem cell reports *3*, 735-742.

5. Zhou, Y., Bond, A.M., Shade, J.E., Zhu, Y., Davis, C.O., Wang, X., Su, Y., Yoon, K.J., Phan, A.T., Chen, W.J., et al. (2018). Autocrine Mfge8 Signaling Prevents Developmental Exhaustion of the Adult Neural Stem Cell Pool. Cell Stem Cell *23*, 444-452 e444.

6. Urban, N., van den Berg, D.L., Forget, A., Andersen, J., Demmers, J.A., Hunt, C., Ayrault, O., and Guillemot, F. (2016). Return to quiescence of mouse neural stem cells by degradation of a proactivation protein. Science *353*, 292-295.